# DISTRIBUTION-INFORMED ONLINE CONFORMAL PREDICTION

**Dongjian Hu**[1][*], **Junxi Wu**[1,2][*], **Shu-Tao Xia**[2][†], **Changliang Zou**[1][†]

[1] School of Statistics and Data Science, LPMC, KLMDASR and LEBPS, Nankai University
[2] Tsinghua Shenzhen International Graduate School, Tsinghua University

## ABSTRACT

Conformal prediction provides a pivotal and flexible technique for uncertainty quantification by constructing prediction sets with a predefined coverage rate. Many online conformal prediction methods have been developed to address data distribution shifts in fully adversarial environments, resulting in overly conservative prediction sets. We propose Conformal Optimistic Prediction (COP), an online conformal prediction algorithm incorporating underlying data pattern into the update rule. Through estimated cumulative distribution function of non-conformity scores, COP produces tighter prediction sets when predictable pattern exists, while retaining valid coverage guarantees even when estimates are inaccurate. We establish a joint bound on coverage and regret, which further confirms the validity of our approach. We also prove that COP achieves distribution-free, finite-sample coverage under arbitrary learning rates and can converge when scores are $i.i.d.$. The experimental results also show that COP can achieve valid coverage and construct shorter prediction intervals than other baselines.

## 1 INTRODUCTION

Uncertainty quantification is essential for making reliable forecasts, as prediction errors could lead to severe consequences, particularly in high-risk domains including epidemiology and finance. Despite the availability of various methods for uncertainty quantification, such as confidence calibration (Guo et al., 2017) and Bayesian networks (Fortunato et al., 2017), these approaches often fail to provide provable coverage guarantees, which significantly restricts their applicability and reliability.

To address this limitation, Conformal Prediction (CP) (Vovk et al., 2005) stands out as a powerful technique to construct prediction sets with model-agnostic and distribution-free properties. Under the assumption of data exchangeability, it theoretically guarantees that the prediction sets contain the true label with a specified probability. Consequently, CP has shown promising performance in multiple domains, such as large language models (Gui et al., 2024), robotics (Lindemann et al., 2023), image classification (Angelopoulos et al., 2020), and decision-making (Bao et al., 2025).

Despite the success of CP for exchangeable data, it remains challenging to directly perform CP in scenarios involving distribution shifts and temporal dependence that violate the exchangeability assumption. To resolve this problem, significant efforts have been made to extend CP methods beyond exchangeability (Barber et al., 2023; Xu & Xie, 2021; Tibshirani et al., 2019; Yang et al., 2024).

Recently, a large amount of research is emerging in Conformal Prediction for time series. Pioneered by ACI (Gibbs & Candès, 2021), adversarial online learning methods were incorporated into the CP framework to adapt to arbitrary online distribution shifts. To improve it, Conformal PID (Angelopoulos et al., 2023b) incorporated PID control and ECI (Wu et al., 2025) leveraged error quantification via smoothed quantile loss. To further determine the step size for dynamic environments, other online learning techniques were used such as directly choosing decaying step size (Angelopoulos et al., 2024), expert aggregation (Zaffran et al., 2022; Gibbs & Candès, 2024), and strongly adaptive regret (Bhatnagar et al., 2023; Hajihashemi & Shen, 2024). However, they focus on fully adversarial environments and adjust prediction sets simply based on the iterative steps, without considering the

---

[*]Equal contribution.
[†]Corresponding authors.

underlying data patterns. Consequently, despite their well-established theoretical basis, they tend to construct overly conservative prediction sets in practice.

Instead of online learning methods, another line of work (Xu & Xie, 2021; 2023; Lee et al., 2024) proposed methods to directly obtain prediction sets via estimated distribution function of non-conformity scores, such as SPCI (Xu & Xie, 2023), which employed random forest as an estimator. Their theoretical results require assumptions like model consistency and distributional smoothness, and hence are not distribution-free. Other methods focus on assigning weights to historical scores. HOP-CPT (Auer et al., 2023) utilized the Modern Hopfield Network and CT-SSF (Chen et al., 2024a) leveraged the inductive bias in deep representation learning to dynamically adjust weights. However, these approaches often lack theoretical results, such as finite-sample coverage, making them potentially unreliable and suffer from overfitting issues.

In this work, we propose *Conformal Optimistic Prediction (COP)*, an online CP algorithm that incorporates underlying data pattern of the non-conformity scores into the update rule. The key step is a refinement update based on the linear approximation of the expected quantile loss and an estimated cumulative distribution function (CDF). COP maintains the well-grounded coverage guarantees inherent to traditional methods, while taking advantage of the distribution information to construct tighter prediction intervals. We also establish a connection to the online optimistic gradient descent (Rakhlin & Sridharan, 2013; Zhao et al., 2024), with a scaled distribution-informed hint. From this standpoint, we derive a joint bound on regret and coverage, which further confirms the validity of our approach. Moreover, COP does not need to compute inverse of the estimated CDF as Lee et al. (2024) and Wang et al. (2025) do, thereby reducing computational cost and unnecessary numerical error. Our contributions can be summarized as follows:

- We propose a novel method for online CP, Conformal Optimistic Prediction (COP). Compared to traditional methods, COP introduces a refinement step leveraging the estimated distribution function, which allows for capturing potential predictable information of the non-conformity scores and providing more efficient prediction sets.

- We also introduce another perspective on COP through the lens of optimistic online gradient descent, which facilitates more comprehensive theoretical analysis. We further explore a joint regret–coverage bound for COP that sheds some light on the theoretical motivation behind our approach.

- We prove the distribution-free and finite-sample coverage of COP with general optimistic terms under arbitrary learning rates. Moreover, we demonstrate the asymptotic consistency of COP with i.i.d. scores and properly chosen learning rates.

- Our experimental results demonstrate the superior performance of COP in time series datasets, including simulation data under distribution shift and real-world data in the finance, energy, and climate domains. We show that COP maintains coverage at the target level and obtains tighter prediction sets than other state-of-the-art methods.

## 2 BACKGROUND

### 2.1 PROBLEM SETUP

Given sequentially collected data $\{(X_t, Y_t)\}_{t \geq 1} \subset \mathcal{X} \times \mathcal{Y}$, at time $t$, our base model $\hat{f}_t$ utilizes previously observed data $\{(X_i, Y_i)\}_{i \leq t-1}$ to produce point prediction $\hat{Y}_t = \hat{f}_t(X_t)$ for the unobserved label $Y_t$. The goal of CP is to construct a prediction set $\hat{C}_t(X_t)$ based on $\hat{Y}_t$. Aligned with the standard CP methods, consider a conformal score function $S_t(\cdot, \cdot) : \mathcal{X} \times \mathcal{Y} \to \mathbb{R}$, which measures the discrepancy between the base model's prediction $\hat{f}(X_t)$ and the true label $Y_t$. For example, the score function can be $S_t(x, y) = |y - \hat{f}_t(x)|$. Given $S_t$, CP constructs the set in the form of:

$$\hat{C}_t(X_t) = \{y \in \mathcal{Y} : S_t(X_t, y) \leq q_t\},$$

where $q_t$ is the threshold to be determined.

Given nominal miscoverage level $\alpha$, note that if the data sequence $\{(X_i, Y_i)\}_{i \leq t+1}$ are i.i.d. or exchangeable, taking $q_t$ to be the $\lceil (1 - \alpha)(1 + t) \rceil$-th smallest elements among $\{S_t(X_t, Y_t)\}_{i=1}^{t}$

yields a valid coverage guarantee by split conformal prediction

$$\mathbb{P}\{Y_{t+1} \in \hat{C}_{t+1}(X_{t+1})\} \geq 1 - \alpha.$$

However, in an adversarial setting where exchangeability does not hold, such as time series data with strong correlations, it is very difficult to achieve such a real-time coverage guarantee. Recent works (Bhatnagar et al., 2023; Angelopoulos et al., 2023b; 2024; Podkopaev et al., 2024) have turned to consider the following long-term coverage guarantee:

$$\lim_{T \to \infty} \frac{1}{T} \sum_{t=1}^{T} \mathbb{1}\{Y_t \notin \hat{C}_t(X_t)\} = \alpha. \tag{1}$$

Parallel to these works, our target lies in designing an online algorithm to dynamically choose the threshold $q_t$ that can achieve the control in equation 1 without any distributional assumptions on data $\{(X_t, Y_t)\}_{t \geq 1}$. Also, we want the size of our prediction sets to be as small as possible.

## 2.2 Conformal Prediction for Sequential Data

Let $s_t = S(X_t, Y_t)$ be the non-conformity score at time $t$. When the label $Y_t$ is observed, online CP considers updating $\hat{q}_t$ with the following rule

$$\hat{q}_{t+1} = \hat{q}_t + \eta(\text{err}_t - \alpha), \tag{2}$$

where $\text{err}_t = \mathbb{1}\{s_t > \hat{q}_t\} = \mathbb{1}\{Y_t \notin \hat{C}_t(X_t)\}$ is the miscoverage indicator and $\eta$ is the learning rate.

The idea to set $q_t$ through online gradient descent with a fixed step size was first introduced in ACI (Gibbs & Candès, 2021). Subsequent improvements include improving update rules by introducing decaying stepsizes (Angelopoulos et al., 2024), PID control (Angelopoulos et al., 2023b), and error quantification (Wu et al., 2025). Moreover, some methods borrow online learning techniques and consider not only coverage guarantee, but also dynamic regret analysis (Gibbs & Candès, 2024; Bhatnagar et al., 2023; Podkopaev et al., 2024). Bao et al. (2024) investigated online selective conformal inference problem for i.i.d. data stream. However, these methods only targeted at fully adversarial environments and ignored the predictive part in the sequence. Another line of work aims at learning the distribution of data with estimated distribution functions (Xu & Xie, 2023; Lee et al., 2024) or capturing temporal structure through deep neural networks (Chen et al., 2024a; Auer et al., 2023). However, both of them cannot provide provable coverage guarantees.

Most relate to us are the LQT algorithm proposed by Areces et al. (2025) and the Conformal PID algorithm in Angelopoulos et al. (2023a). They consider both adversarial and predictable data. However, the application of LQT relies heavily on linear autoregressive structures. For Conformal PID, the selection of its scorecaster model is arbitrary and lacks principled guidance. An inaccurate or excessively complex scorecaster may impair performance by introducing additional variance.

## 2.3 Optimistic Online Gradient Descent

Optimistic online gradient descent (OOGD) has been extensively studied and applied across various domains. It aims at adding prior knowledge about the sequence within the paradigm of online learning. Rakhlin & Sridharan (2013) proposed the general version of the Optimistic Mirror Descent algorithm, which was bulit upon the results in (Chiang et al., 2012). Recently, OOGD has been adopted in Riemannian online convex optimization (Wang et al., 2023; Roux et al., 2025), generative models (Daskalakis et al., 2017), reinforcement learning (Ito, 2021; Bubeck et al., 2019), etc. Other theoretical results of OOGD include problem-dependent regret bounds (Chen et al., 2024b; Zhao et al., 2024), saddle point problem (Daskalakis & Panageas, 2018; Mertikopoulos et al., 2018; Mokhtari et al., 2020) and other last iterate convergence properties (Gorbunov et al., 2022).

## 3 Method and Theory

### 3.1 Conformal Optimistic Prediction

We begin by considering eq. (2) from the view of online graient descent. Let $\ell_{1-\alpha}(q) = (\mathbb{1}\{q > 0\} - \alpha)q$ denote the $(1 - \alpha)$-th quantile loss, then eq. (2) can be viewed as online (sub)gradient

descent (OGD) on the quantile loss $\ell_{1-\alpha}(s_t - q)$. For simplicity, we denote $\nabla_{\hat{q}_t} \ell_{1-\alpha}(s_t - \hat{q}_t)$ as $\nabla \ell_{1-\alpha}(s_t - \hat{q}_t)$, and rewrite eq. (2) as:

$$\hat{q}_{t+1} = \hat{q}_t + \eta(err_t - \alpha) = \hat{q}_t - \eta \nabla \ell_{1-\alpha}(s_t - \hat{q}_t). \tag{3}$$

One critism of eq. (3) is that the good coverage is obtained due to reactively correcting past mistakes and through a cancellation of positive and negative errors. The radius $q_t$ may exhibit significant fluctuations around the true value (Gibbs & Candès, 2021; Wu et al., 2025), leading to inefficient prediction intervals. An alternative way is to directly estimate the conditional quantiles or distribution function $\hat{F}_t$ of the scores in an online fashion (Lee et al., 2024). Then $q_t$ can directly take $\hat{F}_t^{-1}(1-\alpha)$. However, the validity of these methods fully counts on the estimated quantiles or CDF, which may suffer from overfitting when addressing adversarial data.

To mitigate the limitations, we consider improving eq. (3) by taking advantage of the predictable information through an estimated distribution function. Let $\mathcal{S}_t = (s_1, \cdots, s_t)$ and we denote the cumulative distribution function (CDF) of $s_{t+1}$ conditional on $\mathcal{S}_t$ as $F_{t+1}(\cdot|\mathcal{S}_t) = F_{t+1}(\cdot)$ for simplicity. Note that, if the conditional distribution of $s_t|\mathcal{S}_{t-1}$ is invariant over $t$, the target of conformal prediction can be approximately viewed as finding the conditional $(1-\alpha)$-th quantile of $s_t|\mathcal{S}_{t-1}$. Hence, an intuitive way to improve the efficiency of OGD is to adopt the following refinement for $\hat{q}_{t+1}$. For some $\lambda_{t+1} > 0$:

$$q_{t+1} = \arg\min_q \mathbb{E}_{s_{t+1}}\left[\ell_{1-\alpha}(s_{t+1} - q)|\mathcal{S}_t\right] + \frac{1}{2\lambda_{t+1}}\|q - \hat{q}_{t+1}\|_2^2.$$

Given that we output $q_t$ at timestep $t$, the miscoverage indicator of interest turns into $err_t = \mathbb{1}\{s_t > q_t\}$, instead of $\mathbb{1}\{s_t > \hat{q}_t\}$ in eq. (3). Hence, eq. (3) is substituted with:

$$\hat{q}_{t+1} = \hat{q}_t + \eta(err_t - \alpha) = \hat{q}_t - \eta \nabla \ell_{1-\alpha}(s_t - q_t). \tag{4}$$

The refinement serves as a re-calibration step to adjust and optimize the conservative action that update eq. (4) makes, and utilizes the predictable information of non-conformity scores $\{s_t\}_{t\geq 1}$ to enhance the conditional performance.

Let $\mathcal{L}_{t+1}(q) = \mathbb{E}_{s_{t+1}}[\ell_{1-\alpha}(s_{t+1} - q)|\mathcal{S}_t]$. Note that $\nabla_q \mathcal{L}_{t+1}(q) = F_{t+1}(q) - (1-\alpha)$, the optimization above does not have a closed form solution. Similar issues also occur when using online mirror descent. We discuss this in Appendix A.1. As opposed to the above implicit update, we therefore turn to minimize the local approximation.

By convexity of $\mathcal{L}_{t+1}$:

$$\mathcal{L}_{t+1}(q) \geq \mathcal{L}_{t+1}(\hat{q}_{t+1}) + (q - \hat{q}_{t+1})(F_{t+1}(\hat{q}_{t+1}) - (1-\alpha))$$

Denote $\hat{F}_{t+1}$ as the estimated CDF of $s_{t+1}|\mathcal{S}_t$. Replacing $F_{t+1}$ with $\hat{F}_{t+1}$, the optimization above reduces to:

$$q_{t+1} = \arg\min_q \mathcal{L}_{t+1}(\hat{q}_{t+1}) + (q - \hat{q}_{t+1})\left(\hat{F}_{t+1}(\hat{q}_{t+1}) - (1-\alpha)\right) + \frac{1}{2\lambda_{t+1}}\|q - \hat{q}_{t+1}\|_2^2.$$

Equivalently, we have

$$q_{t+1} = \hat{q}_{t+1} - \lambda_{t+1}\left(\hat{F}_{t+1}(\hat{q}_{t+1}) - (1-\alpha)\right). \tag{5}$$

We remark that our refinement can also be implemented on another widely employed online CP method, adaptive conformal inference (ACI) (Gibbs & Candès, 2021), as detailed in Appendix A.2. There are many choices for $\hat{F}_{t+1}$, such as empirical cumulative distribution function (ECDF) and kernel-based estimation (Silverman, 1986). As a brief clarification on the benefit of our refinement, we present the proposition as follows.

**Proposition 1.** *Assume that $\hat{F}_{t+1}(\hat{q}_{t+1}) - (1-\alpha)$ and $F_{t+1}(\hat{q}_{t+1}) - (1-\alpha)$ have the same sign, and $F_{t+1}$ is L-Lipschitz continuous. With a suitably small $\lambda_{t+1} > 0$, we have:*

$$\mathbb{E}\left[\ell_{1-\alpha}(s_{t+1} - q_{t+1})|\mathcal{S}_t\right] \leq \mathbb{E}\left[\ell_{1-\alpha}(s_{t+1} - \hat{q}_{t+1})|\mathcal{S}_t\right],$$

i.e. the expected quantile loss of $q_{t+1}$ is smaller than that of $\hat{q}_{t+1}$ at timestep $t+1$. This indicates that we can tolerate a certain deviation of $\hat{F}_{t+1}$ from $F_{t+1}$. It suffices that their relative ordering with respect to $1 - \alpha$ coincides at certain points and our method is preferable in this regard. We also add a small modification of eq. (5) in Appendix A.3 that avoids the "same-sign" assumption. Moreover, even if the estimated distribution function $\hat{F}_{t+1}$ is not accurate, we can still obtain the basic long-term coverage guarantee, as shown in Proposition 2.

Formally, we propose our method Conformal Optimistic Prediction (COP) in Algorithm 1. The naming reflects the "optimistic" belief that the estimated CDF $\hat{F}_{t+1}$ reflects the behavior of the true CDF $F_{t+1}$. We will further clarify the close connection between our method and online optimistic gradient descent (OOGD) in the next section.

---

**Algorithm 1** Conformal Optimistic Prediction with estimated CDF

---

**Require:** nominal miscoverage rate $\alpha \in (0,1)$, base predictor $\hat{f}$, learning rate $\eta > 0, t = 1, 2, \cdots, T$, scale factor $\lambda_{t+1} \geq 0$, init. $\hat{q}_1 \in \mathbb{R}$

1: **for** $t \geq 1$ **do**
2:     Observe input $X_t \in \mathcal{X}$
3:     Return prediction set $\hat{C}_t(X_t, q_t) = [\hat{f}_t(X_t) - q_t, \hat{f}_t(X_t) + q_t]$
4:     Observe true label $Y_t \in \mathcal{Y}$ and compute true radius $s_t = \inf\{s \in \mathbb{R} : Y_t \in \hat{C}_t(X_t, s)\}$
5:     Update the primary radius

$$\hat{q}_{t+1} = \hat{q}_t + \eta \left[ \mathbb{1}(s_t > q_t) - \alpha \right]$$

6:     Compute estimated cumulative distribution function $\hat{F}_{t+1}$ for $s_{t+1}$
7:     Update the refined radius

$$q_{t+1} = \hat{q}_{t+1} - \lambda_{t+1} \left( \hat{F}_{t+1}(\hat{q}_{t+1}) - (1 - \alpha) \right)$$

8: **end for**

---

## 3.2 CONNECTION WITH OOGD

To begin, we note that eq. (4) and eq. (5) can be rewritten in the way of OOGD by:

$$\widehat{q}_{t+1} = \arg\min_q \left\{ \eta \left\langle \nabla \ell_{1-\alpha}(s_t - q_t), q \right\rangle + \frac{1}{2} \|q - \widehat{q}_t\|^2 \right\} \tag{6}$$

$$q_{t+1} = \arg\min_q \left\{ \eta \left\langle M_{t+1}, q \right\rangle + \frac{1}{2} \|q - \widehat{q}_{t+1}\|^2 \right\}. \tag{7}$$

In the literature of OOGD, $M_{t+1}$ is the optimistic term, aiming at guessing the next move and incorporating it into the objective (Rakhlin & Sridharan, 2013). While classical OOGD often takes $M_{t+1}$ as the previous gradient, COP uses a distribution-informed hint:

$$M_{t+1} = (\hat{F}_{t+1}(\hat{q}_{t+1}) - (1 - \alpha)) \cdot \lambda_{t+1}/\eta, \tag{8}$$

which captures potential distribution shift of the next score. Empirically, we take $\lambda_{t+1}/\eta \leq 1$ and denote it as the scale factor, indicating our confidence on the accuracy of $\hat{F}_{t+1}$. For instance, when the base model performs well, the scores $s_t$ tend to be highly stochastic and difficult to predict, suggesting a smaller $\lambda_{t+1}$. Conversely, when $\hat{F}_t$ is reliable, the scale factor can be set close to 1. Specifically, COP boils down to OGD if we simply set $\lambda_{t+1} = 0$.

Besides the coverage guarantee in Section 3.3, dynamic regret is often considered as an additional performance metric for online CP (Bhatnagar et al., 2023; Ramalingam et al., 2025). From the view of OOGD, we obtain a joint regret-coverage bound with constant learning rate in Theorem 1. The result for dynamic learning rates can be seen in Appendix B.2.

**Theorem 1.** *Let $\ell_t(q) = \ell_{1-\alpha}(s_t - q)$. For arbitrary $\{u_t\}_{t \geq 1}$, $u_0 = 0$, we have:*

$$\underbrace{\frac{1}{T}\sum_{t=1}^{T}[\ell_t(q_t) - \ell_t(u_t)]}_{regret} + \underbrace{\frac{\eta(1-2\alpha)}{4}\left(\frac{\sum_{t=1}^{T} err_t}{T} - \alpha\right)}_{coverage} \leq \frac{\eta}{T}\sum_{t=1}^{T}\|\alpha - err_t - M_t\|_2^2$$

$$+ \underbrace{\sum_{t=1}^{T}\frac{1}{2\eta}\left(\|u_t - \hat{q}_t\|_2^2 - \|u_{t-1} - \hat{q}_t\|_2^2\right)}_{environments}.$$

The term $\sum_{t=1}^{T}\frac{1}{2\eta}\left(\|u_t - \hat{q}_t\|_2^2 - \|u_{t-1} - \hat{q}_t\|_2^2\right)$ reflects the non-stationarity of environments (Zhao et al., 2024). Although without distributional assumption, regret bounds and coverage bounds of online CP do not imply each other (Angelopoulos et al., 2025), Theorem 1 shows that properly choosing $M_t$ can simultaneously reduce the bound for both regret and coverage. Specifically, note that $\mathbb{E}\|\alpha - err_t - M_t\|_2^2|\mathcal{S}_{t-1} \approx \mathbb{E}\|\alpha - \mathbb{1}(s_t > \hat{q}_t) - M_t\|_2^2|\mathcal{S}_{t-1}$, a natural idea for sequentially choosing $M_t$ would be taking it close to:

$$F_t(\hat{q}_t) - (1 - \alpha) = \arg\min_{M_t}\mathbb{E}\|\alpha - \mathbb{1}(s_t > \hat{q}_t) - M_t\|_2^2|\mathcal{S}_{t-1},$$

This coincides with the choice of our method with the scale factor taken as 1.

### 3.3 COVERAGE GUARANTEES

In this section, we present coverage guarantees of COP. For simplicity, we consider the refinement step eq. (5) in the way of OOGD:

$$q_{t+1} = \hat{q}_{t+1} - \eta_t M_{t+1}, \tag{9}$$

where $M_{t+1}$ is the optimistic term, instead of the choice in Equation (8). All of the following results require $M_{t+1}$ to be bounded, and this holds in our case since the scale factor $\lambda_{t+1}/\eta_t \leq 1$ and $M_t \in [\alpha - 1, \alpha]$ in Equation (8).

The detailed proofs can be found in Appendix B. We first show the long term coverage of COP with fixed learning rate $\eta$.

**Proposition 2.** *Assume that for each $t$, $s_t \in [0, B]$ and $M_t \in [-M, M]$, then for all $T \geq 1$, the prediction sets generated by COP with fixed learning rate $\eta$ satisfies*

$$\left|\frac{1}{T}\sum_{t=1}^{T}err_t - \alpha\right| \leq \frac{B + (2 + 6M)\eta}{T\eta}. \tag{10}$$

Although we assume boundedness of scores $s_t$ and optimistic terms $M_t$, running COP and achieving eq. (10) does not require knowing the upper bound $B$ and $M$. The finite-sample result in Proposition 2 naturally implies the long-term coverage eq. (1), *i.e.* $\lim_{T \to \infty}\frac{\sum_{t=1}^{T} err_t}{T} = \alpha$.

Besides, we further develop coverage guarantee with dynamic learning rate $\eta_t$:

**Theorem 2.** *Assume that for each $t$, $s_t \in [0, B]$ and $M_t \in [-M, M]$, then for all $T \geq 1$, the prediction sets generated by COP with arbitrary learning rate $\eta_t$ satisfies*

$$\left|\frac{1}{T}\sum_{t=1}^{T}err_t - \alpha\right| \leq \frac{B + (2 + 6M)\Omega_T}{T}\|\Delta_{1:T}\|_1,$$

*where $\Delta_1 = \eta_1^{-1}, \Delta_t = \eta_t^{-1} - \eta_{t-1}^{-1}$ for all $t \geq 1$, and $\|\Delta_{1:T}\|_1 = \sum_{t=1}^{T}\Delta_t, \Omega_T = \max_{1 \leq r \leq T}\eta_r$.*

Note that, $\|\Delta_{1:T}\|_1$ is closely related to the number of times we increase the step size. Specifically, $\|\Delta_{1:T}\|_1 \leq 2N_t/(\min_{t \leq T}\eta_t)$, where $N_T = \sum_{t=1}^{T}\mathbb{1}(\eta_{t+1} > \eta_t)$. Hence, the upper bound in Theorem 2 will converge to zero as long as $N_T/(\min_{t \leq T}\eta_t) = o(T)$. In particular, if we consider decaying $\eta_t$ in (Angelopoulos et al., 2024), then $\Omega_T = \eta_1, \|\Delta_{1:T}\|_1 = 1/\eta_T, \|\Delta_{1:T}\|_1/T \leq N_T/T\eta_T$. Setting $\eta_T = o(N_T/T)$ ensures the long-term coverage in eq. (1).

In the following part, we derive the asymptotic coverage property of COP under appropriately chosen learning rates. For this, we require that the scores are *i.i.d.*.

**Theorem 3.** *Assume that the scores $s_t \overset{iid}{\sim} P$ and has a continuous distribution function $F$. The learning rates $\{\eta_t\}$ satisfy:*

$$\sum_{t=1}^{\infty} \eta_t = \infty, \; \sum_{t=1}^{\infty} \eta_t^2 < \infty.$$

*Let $q^*$ be the $(1-\alpha)$-th quantile of $P$, satisfying: for $q > q^*$, $F(q) > 1 - \alpha$ and for $q < q^*$, $F(q) < 1 - \alpha$, Then $q_t \to q^*$, i.e. $\lim_{t\to\infty} P(Y_t \in C_t(X_t)) = 1 - \alpha$.*

Choosing constant learning rates in eq. (3) will cause oscillations (Gibbs & Candès, 2024). Theorem 3 demonstrates that, under the i.i.d. assmptions and choosing learning rates satisfying the Robbins-Monro condition (Robbins & Monro, 1951), $q_t$ exhibits a convergence behavior, even if we add bounded optimistic terms $M_t$ in every update.

## 4 EXPERIMENTS

### 4.1 SETUP

**Datasets.** We evaluate five simulation datasets under changepoints, distribution drift (Barber et al., 2023), variance changepoint, heavy-tailed noise, and extreme distribution drift. Besides, we evaluate four real-world datasets: Amazon stock, Google stock (Nguyen, 2018), electricity demand (Harries et al., 1999) and temperature in Delhi (Vrao., 2017). In the subsequent sections, we will provide a detailed introduction to each of these datasets.

**Base predictors.** We evaluate three base predictors that have distinct levels of forecasting performance. The Prophet model, a Bayesian additive model, forecasts $\hat{Y}_t = g(t) + s(t) + h(t) + \epsilon_t$, capturing the overall trend, seasonality, holidays, and noise. The AR model forecasts $\hat{Y}_t = \phi_1 Y_{t-1} + \dots + \phi_p Y_{t-p} + \epsilon_t$, with $p = 3$. The Theta model decomposes the series by adjusting the curvature with coefficients $\theta = 0$ (long-term trend) and $\theta = 2$ (short-term dynamics).

**Baselines.** We compare with seven state-of-the-art methods: ACI (Gibbs & Candès, 2021), OGD, SF-OGD (Bhatnagar et al., 2023), decay-OGD Angelopoulos et al. (2024), Conformal PID (Angelopoulos et al., 2023b), ECI (Wu et al., 2025) and LQT(fixed) (Areces et al., 2025). Several follow-up works on ACI consider adaptively setting learning rates via the expert aggregation technique (Gibbs & Candès, 2024; Bhatnagar et al., 2023). COP and the seven baselines are orthogonal to these works and can be naturally incorporated by serving as a single expert.

**General implements.** We choose the target coverage $1 - \alpha = 90\%$ and construct asymmetric prediction sets using two-side quantile scores under $\alpha/2$ respectively. For prediction sets, all baselines will output asymmetric sets $[\hat{Y}_t - q_t^l, \hat{Y}_t + q_t^u]$ with upper score $q_t^u$ and lower score $q_t^l$ under half of the coverage level $\alpha/2$ respectively.

**Hyperparameters.** The proposed COP has three hyperparameters, the base learning rate $\eta$, scale factor $\lambda = 0.5$ and the window length $w = 100$. Same as previous works, the adaptive learning rates $\eta_t = \eta \cdot (\max\{s_{t-w+1}, \dots, s_t\} - \min\{s_{t-w+1}, \dots, s_t\})$. For reproducibility, all baseline implementations leverage the open-source Python codes from Wu et al. (2025) or Areces et al. (2025). The hyperparameters of LQT need to be tuned via grid search. The operational ranges of $\eta$ and more details about $\eta_t$ for each methods can be found in Appendix D.

**Choices of estimated CDF.** By default, we set the estimated CDF as empirical CDF, which is $\hat{F}_{t+1}(\hat{q}_{t+1}) = \frac{1}{w} \sum_{i=t-w+1}^{t} \mathbb{1}\{s_i \leq \hat{q}_{t+1}\}$. In addition, we conduct experiments with estimated CDF based on the kernel density estimator in Appendix C.

**Evaluation metrics.** The coverage rate measures the proportion of time steps where the true observation $Y_t$ falls within the prediction set $C_t(X_t, q_t)$. The width of the prediction set reflect the efficiency of CP, including the average width (reflecting overall performance) and the median width (robust to outliers and extreme intervals). A well-calibrated method should achieve coverage close to the predefined target level, while keeping the width as short as possible. We also evaluate recovery time in Appendix F, statistical significance in Appendix I, and average time cost in Appendix J. For visualization, we also provide the figures in Appendix K.

**Overview of experimental results.** We have conducted extensive experiments, including five simulation datasets in Section 4.2 and Appendix E, four real-world datasets in Section 4.3. We have also conducted some ablation studies, including the choice of estimated CDF in Appendix C and the scale factor in Appendix H. To evaluate the case that estimated CDF is inaccurate, we have conducted the inaccurate estimated CDF in Appendix G. Our code is available at `https://github.com/creator-xi/Conformal-Optimistic-Prediction`.

## 4.2 SIMULATION DATASET

We evaluated our method on two simulation datasets under changepoints and distribution drift setting, respectively. Both datasets $\{X_i, Y_i\}_{i=1}^n$ are generated according to a linear model $Y_t = X_t^T \beta_t + \epsilon_t$, $X_t \sim \mathcal{N}(0, I_4)$, $\epsilon_t \sim \mathcal{N}(0, 1)$, $n = 2000$.

- Changepoint setting: we set two changepoints: $\beta_t = \beta^{(0)} = (2, 1, 0, 0)^\top$ for $t = 1, \ldots, 500$; $\beta_t = \beta^{(1)} = (0, -2, -1, 0)^\top$ for $t = 501, \ldots, 1500$; and $\beta_t = \beta^{(2)} = (0, 0, 2, 1)^\top$ for $t = 1501, \ldots, 2000$.
- Distribution drift setting: we set $\beta_1 = (2, 1, 0, 0)^\top$, $\beta_n = (0, 0, 2, 1)^\top$, and use linear interpolation to compute $\beta_t = \beta_1 + \frac{t-1}{n-1}(\beta_n - \beta_1)$.

Table 1: The experimental results in the two simulation datasets with nominal level $\alpha = 10\%$.

| Dataset | Method | Prophet | | | AR | | | Theta | | |
|---|---|---|---|---|---|---|---|---|---|---|
| | | Coverage (%) | Average width | Median width | Coverage (%) | Average width | Median width | Coverage (%) | Average width | Median width |
| Changepoint | ACI | 89.9 | $\infty$ | 8.20 | 89.9 | $\infty$ | 8.20 | 89.9 | $\infty$ | 8.43 |
| | OGD | 90.0 | 8.49 | 8.50 | 89.9 | 8.39 | 8.40 | 89.9 | 8.73 | 8.70 |
| | SF-OGD | 90.0 | 12.48 | 11.56 | 90.0 | 12.58 | 11.69 | 89.9 | 12.70 | 11.88 |
| | decay-OGD | 90.0 | 8.30 | **8.22** | 90.0 | 8.26 | 8.21 | 90.0 | 8.57 | 8.60 |
| | PID | 89.7 | 11.02 | 9.64 | 89.9 | 10.83 | 9.35 | 89.7 | 11.23 | 9.78 |
| | ECI | 89.9 | **8.16** | 8.25 | 89.9 | **8.17** | 8.26 | 89.8 | 8.55 | 8.68 |
| | LQT | 89.8 | 8.49 | 8.31 | 89.6 | 8.54 | 8.29 | 89.8 | 9.29 | 8.75 |
| | COP | 89.8 | 8.29 | 8.44 | 89.7 | 8.18 | **8.25** | 89.8 | **8.45** | **8.53** |
| Distribution Drift | ACI | 89.9 | $\infty$ | 6.69 | 89.8 | $\infty$ | 6.56 | 89.9 | $\infty$ | 6.79 |
| | OGD | 90.3 | 7.24 | 7.05 | 90.2 | 7.15 | 7.10 | 90.3 | 7.34 | 7.25 |
| | SF-OGD | 90.0 | 11.48 | 10.34 | 89.9 | 11.46 | 10.31 | 89.9 | 11.95 | 10.54 |
| | decay-OGD | 90.6 | 7.64 | 6.95 | 90.4 | 7.31 | **6.81** | 90.6 | 7.62 | **6.91** |
| | PID | 89.7 | 9.41 | 7.81 | 89.8 | 10.08 | 7.92 | 89.7 | 10.06 | 7.90 |
| | ECI | 90.0 | 7.27 | 6.98 | 90.0 | **7.06** | 6.98 | 90.2 | 7.55 | 7.18 |
| | LQT | 90.6 | 9.74 | 8.72 | 91.9 | 8.16 | 7.22 | 91.0 | 10.48 | 8.86 |
| | COP | 90.6 | **7.07** | **6.89** | 90.0 | 7.09 | 6.97 | 90.9 | **7.30** | 6.99 |

The quantitative results are shown in Table 1. ACI frequently produces infinitely wide prediction sets due to updating $\alpha_t$. The overly conservative sets undermine the utility of prediction. OGD and SF-OGD partially balance coverage and width, but their performance is overly sensitive to learning rates. In contrast, decay-OGD performs better in terms of median width due to the stability of decaying learning rate in the later stages. Conformal PID borrows from PID control for adjustment and needs to train scorecasters. LQT is sensitive to hyperparameters and relies on grid search, which makes its performance unstable. ECI reacts quickly to distribution shifts through error-quantification, and hence achieves relatively tight sets. However, ECI may struggle in complex data environments since it cannot capture the underlying information of data.

As for COP, it maintains the coverage rate close to the nominal $90\%$ level and tighter widths than other methods. COP achieves this by incorporating an optimistic term based on the estimated cumulative distribution function, which preserves the long-term coverage guarantee of traditional online CP while using predictable distribution information to adjust the width more precisely. The experimental results also show the generality to adapt to different base predictors across Prophet, AR, and Theta models.

### 4.3 REAL-WORLD DATASETS

Moreover, we evaluated our method on four real-world time series datasets across three critical domains. All datasets retain raw temporal ordering to preserve real-world sequential dependencies.

- Financial markets: daily opening prices of Amazon and Google from 2006 to 2014, capturing non-stationary trends, regime shifts like the 2008 crisis, and heteroskedastic volatility. The base predictors will forecast the daily opening price on a log scale.
- Energy systems: New South Wales electricity demand for half an hour from 1996 to 1998 normalized to $[0, 1]$, featuring multiscale periodicity and demand surges).
- Climate science: daily Delhi temperatures from 2003 to 2017 reflecting seasonal cycles, long-term warming trends, and extreme weather anomalies.

Table 2: The experimental results in the four real-world datasets with nominal level $\alpha = 10\%$.

| Dataset | Method | Prophet | | | AR | | | Theta | | |
|---|---|---|---|---|---|---|---|---|---|---|
| | | Coverage (%) | Average width | Median width | Coverage (%) | Average width | Median width | Coverage (%) | Average width | Median width |
| Amazon Stock | ACI | 90.2 | $\infty$ | 46.97 | 89.8 | $\infty$ | 13.77 | 89.7 | $\infty$ | 12.31 |
| | OGD | 89.6 | 55.15 | 30.00 | 89.9 | 19.10 | 15.00 | 89.8 | 18.07 | 14.50 |
| | SF-OGD | 89.5 | 61.47 | 31.75 | 89.9 | 24.44 | 21.05 | 90.0 | 23.88 | 21.14 |
| | decay-OGD | 89.9 | 97.22 | 36.20 | 89.7 | 20.23 | 14.01 | 89.2 | 17.49 | 13.46 |
| | PID | 89.8 | 52.56 | 39.09 | 89.6 | 59.22 | 37.93 | 89.5 | 61.19 | 40.20 |
| | ECI | 90.1 | 47.00 | 34.84 | 89.5 | 17.12 | 12.73 | 89.7 | 17.46 | 12.49 |
| | LQT | 89.3 | **31.42** | **15.26** | 90.3 | 21.61 | 18.92 | 89.9 | 27.17 | 15.13 |
| | COP | 89.6 | 39.86 | 27.91 | 89.5 | **17.09** | **12.90** | 89.6 | **17.21** | **12.23** |
| Google Stock | ACI | 90.0 | $\infty$ | 66.83 | 89.8 | $\infty$ | 18.64 | 90.5 | $\infty$ | 32.78 |
| | OGD | 89.7 | 57.60 | 46.00 | 90.7 | 33.76 | 23.00 | 89.9 | 31.49 | 29.50 |
| | SF-OGD | 89.6 | 58.92 | 47.78 | 89.9 | 28.31 | 24.42 | 90 | 34.04 | 31.48 |
| | decay-OGD | 89.9 | 77.23 | 50.18 | 90.2 | 46.53 | 26.77 | 90.2 | 55.32 | 33.71 |
| | PID | 90.1 | 57.47 | 48.44 | 89.9 | 64.88 | 54.07 | 89.9 | 63.58 | 54.05 |
| | ECI | 89.9 | 56.06 | 46.96 | 89.7 | 19.95 | 17.19 | 89.6 | 30.92 | 29.53 |
| | LQT | 89.9 | 57.31 | 47.00 | 90.5 | 41.80 | 25.00 | 89.6 | 41.70 | 37.58 |
| | COP | 89.7 | **49.72** | **42.09** | 89.6 | **19.87** | **17.04** | 89.3 | **30.25** | **28.24** |
| Electricity Demand | ACI | 90.1 | $\infty$ | 0.443 | 90.1 | $\infty$ | 0.105 | 90.2 | $\infty$ | **0.055** |
| | OGD | 89.8 | 0.433 | 0.435 | 90.0 | 0.133 | 0.115 | 90.1 | 0.081 | 0.075 |
| | SF-OGD | 89.9 | 0.419 | 0.426 | 90.0 | 0.129 | 0.116 | 90.3 | 0.106 | 0.095 |
| | decay-OGD | 90.1 | 0.531 | 0.521 | 90.1 | 0.122 | 0.099 | 90.0 | 0.100 | 0.059 |
| | PID | 90.1 | **0.207** | **0.177** | 90.0 | 0.434 | 0.432 | 89.9 | 0.413 | 0.411 |
| | ECI | 90.0 | 0.384 | 0.382 | 90.0 | **0.117** | **0.098** | 89.9 | 0.071 | 0.055 |
| | LQT | 90.1 | 0.221 | 0.218 | 90.1 | 0.144 | 0.138 | 90.0 | 0.113 | 0.111 |
| | COP | 90.1 | 0.385 | 0.376 | 90.0 | **0.117** | **0.098** | 89.8 | **0.069** | **0.052** |
| Temperature | ACI | 91.0 | $\infty$ | 8.49 | 90.0 | $\infty$ | 6.06 | 90.2 | $\infty$ | 6.48 |
| | OGD | 90.4 | 7.54 | 7.60 | 90.1 | 6.82 | 6.10 | 90.0 | 6.36 | 6.30 |
| | SF-OGD | 90.0 | 7.17 | 7.08 | 90.1 | 6.37 | 5.91 | 90.1 | 6.75 | 6.43 |
| | decay-OGD | 90.1 | 8.84 | 8.35 | 90.0 | 6.36 | 5.67 | 89.9 | 6.56 | 6.18 |
| | PID | 90.1 | 7.65 | 7.65 | 89.7 | 8.92 | 8.86 | 89.7 | 8.77 | 8.79 |
| | ECI | 90.0 | 7.20 | 7.22 | 90.1 | 6.39 | 6.10 | 90.0 | 6.41 | 6.27 |
| | LQT | 90.2 | 8.57 | 7.30 | 90.3 | 6.78 | 6.00 | 90.1 | 7.51 | 7.08 |
| | COP | 90.1 | **7.05** | **7.07** | 89.9 | **5.85** | **5.58** | 90.0 | **6.27** | **6.18** |

The experimental results under the real-world dataset demonstrate the performance of each method in real-world scenarios. As can be seen from the data in Table 2, COP shows superior performance. ACI generally maintains the coverage rate at nominal level, but its prediction intervals are often infinitely

wide, which makes the results lack practical application value. The OGD series of methods achieve a more balanced trade-off between coverage and interval width, but they are highly sensitive to learning rate selection. Conformal PID trains scorecasters to compensate for the base predictor's accuracy and often improve when the base predictor is less accurate. ECI produces tight prediction sets through error quantification while maintain the coverage rate. Overall, these results highlight the advantages and limitations of each method in real-world applications.

## 5 CONCLUSIONS

In this work, we introduce Conformal Optimistic Prediction (COP), a novel online CP algorithm that leverages estimated cumulative distribution functions of non-conformity scores. Viewing COP through optimistic online gradient descent enables a comprehensive theory, including a joint regret–coverage bound that clarifies its motivation. Theoretically, we also prove distribution-free, finite-sample coverage for general optimistic updates under arbitrary learning rates, and show asymptotic consistency with $i.i.d.$ scores under suitable rates. Experiments on synthetic distribution shift and real time-series data from finance, energy, and climate demonstrate that COP attains target coverage while producing tighter prediction sets than state-of-the-art alternatives.

## ACKNOWLEDGEMENT

We would like to thank all anonymous reviewers and the area chair for their helpful comments. Shu-Tao Xia was supported in part by the National Natural Science Foundation of China under Grant 62571298. Changliang Zou was supported by the National Key R&D Program of China (Grant Nos. 2022YFA1003703, 2022YFA1003800), and the National Natural Science Foundation of China (Grant No. 12231011).

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

# A  SOME DISCUSSIONS

## A.1  DISCUSSIONS ON ONLINE MIRROR DESCENT

We remark that if we can obtain the estimated distribution function $\hat{F}_t$ in an online fashion, a direct way to combine it with eq. (3) is to leverage online mirror descent (Orabona, 2019; Wang et al., 2025):

$$\nabla\psi_{t+1}(q_{t+1}) = \nabla\psi_t(q_t) + \eta_t(err_t - \alpha), \tag{11}$$

where $err_t = \mathbb{1}\{s_t > q_t\}$, $\psi_t(q) = \mathbb{E}_{s_t}[\ell_{1-\alpha}(s_t - q)|\mathcal{S}_{t-1}] + \sigma q^2/2$, and $\sigma > 0$ is a tuning parameter. Equation (11) is equivalent to:

$$F_{t+1}(q_{t+1}) + \sigma q_{t+1} = F_t(q_t) + \sigma q_t + \eta_t(err_t - \alpha). \tag{12}$$

Further, if $s_{t+1}$ has a probability density function (PDF) that is bounded away from zero almost surely, we can directly use:

$$F_{t+1}(q_{t+1}) = F_t(q_t) + \eta_t(err_t - \alpha). \tag{13}$$

It suffices to use $\hat{F}$ to substitute $F$ above. However, both eq. (12) and eq. (13) do not have closed form solutions for $q_{t+1}$, which makes the update difficult to compute.

## A.2  DISCUSSIONS ON ACI

Adaptive Conformal Inference (ACI) (Gibbs & Candès, 2021) is another widely used online CP update besides eq. (3). Let $\hat{Q}_t(\cdot)$ be the fitted quantiles of the non-conformity scores in calibration set $\mathcal{D}_{\text{cal}}$:

$$\hat{Q}(p) := \inf\left\{s : \left(\frac{1}{|\mathcal{D}_{\text{cal}}|}\sum_{(X_r,Y_r)\in\mathcal{D}_{\text{cal}}}\mathbb{1}_{\{S_r(X_r,Y_r)\leq s\}}\right) \geq p\right\}.$$

For prediction set $\hat{C}_t(\alpha) := \{y : S_t(X_t, y) \leq \hat{Q}_t(1-\alpha)\}$, define:

$$\beta_t := \sup\{\beta : Y_t \in \hat{C}_t(\beta)\}.$$

Denote $err_t = \mathbb{1}(\hat{\alpha}_t > \beta_t)$, then ACI follows the iteration:

$$\begin{aligned}
1 - \hat{\alpha}_{t+1} &= 1 - \hat{\alpha}_t + \eta(err_t - \alpha) \\
&= 1 - \hat{\alpha}_t - \eta\nabla\ell_{1-\alpha}(1 - \beta_t - (1 - \hat{\alpha}_t)).
\end{aligned}$$

Note that $1 - \beta_t$ can be viewed as $s_t$ in eq. (3), the similar refinement is as follows:

$$1 - \alpha_{t+1} = 1 - \hat{\alpha}_{t+1} - \lambda_{t+1}\left(\hat{F}_{t+1}(1 - \hat{\alpha}_{t+1}) - (1 - \alpha)\right),$$

i.e.

$$\alpha_{t+1} = \hat{\alpha}_{t+1} + \lambda_{t+1}\left(\hat{F}_{t+1}(1 - \hat{\alpha}_{t+1}) - (1 - \alpha)\right),$$

where $\hat{F}_{t+1}$ is the estimated CDF of $(1 - \beta_{t+1})$. However, $\alpha_t < 0$ or $\alpha_t > 1$ can happen frequently for some $\eta < 0$ and may output infinite or null prediction sets. Hence, we do not adopt this kind of update.

## A.3  DISCUSSIONS ON THE "SAME-SIGN" ASSUMPTION

In order to clarify the superiority of the refinement step, we introduce a "same-sign" assumption in Proposition 1, which is unverifiable in real-world data. However, note that

$$[\hat{F}_{t+1}(\hat{q}_{t+1}) - (1 - \alpha)] \cdot [F_{t+1}(\hat{q}_{t+1}) - (1 - \alpha)] \geq 0$$

follows from

$$|\hat{F}_{t+1}(\hat{q}_{t+1}) - (1 - \alpha)| \geq \sup_q |F_{t+1}(q) - \hat{F}_{t+1}(q)| \triangleq \epsilon_{t+1}.$$

Hence, an intuitive way to avoid the unverifiable assumption is to replace eq. (5) with:

$$q_{t+1} = \hat{q}_{t+1} - \lambda_{t+1} \mathbb{1}(|\hat{F}_{t+1}(\hat{q}_{t+1}) - (1-\alpha)| \geq \epsilon_{t+1}) \left( \hat{F}_{t+1}(\hat{q}_{t+1}) - (1-\alpha) \right). \quad (14)$$

For deployability, $\epsilon_{t+1}$ can be viewed as a hyperparameter that depends on the temporal properties of data and the accuracy of $\hat{F}_{t+1}$. Following the same proof of Proposition 1, we have

$$\mathbb{E}\left[\ell_{1-\alpha}(s_{t+1} - q_{t+1})|\mathcal{S}_t\right] \leq \mathbb{E}\left[\ell_{1-\alpha}(s_{t+1} - \hat{q}_{t+1})|\mathcal{S}_t\right],$$

as long as $F_{t+1}$ is $L$-Lipschitz and $\lambda_{t+1} > 0$ is small.

# B PROOFS

## B.1 PROOF OF PROPOSITION 1

**Proposition 1.** Assume that $\hat{F}_{t+1}(\hat{q}_{t+1}) - (1-\alpha)$ and $F_{t+1}(\hat{q}_{t+1}) - (1-\alpha)$ have the same sign, and $F_{t+1}$ is $L$-Lipschitz. With a suitably small $\lambda > 0$, we have:

$$\mathbb{E}\left[\ell_{1-\alpha}(s_{t+1} - q_{t+1})|\mathcal{S}_t\right] \leq \mathbb{E}\left[\ell_{1-\alpha}(s_{t+1} - \hat{q}_{t+1})|\mathcal{S}_t\right],$$

*Proof.* Let $\mathcal{L}(q) = \mathbb{E}_{s_{t+1}} \ell_{1-\alpha}(s_{t+1} - q)$. Note that the $L$-Lipschitzness of $F_{t+1}$ implies that $\nabla \mathcal{L}(q)$ is $L$-Lipschitz continuous. Hence:

$$\mathcal{L}(q_{t+1}) - \mathcal{L}(\hat{q}_{t+1}) \leq \nabla \mathcal{L}(\hat{q}_{t+1})(q_{t+1} - \hat{q}_{t+1}) + \frac{L}{2}\|q_{t+1} - \hat{q}_{t+1}\|_2^2$$

$$= -\lambda_{t+1}\left[\hat{F}_{t+1}(\hat{q}_{t+1}) - (1-\alpha)\right] \nabla g(\hat{q}_{t+1}) + \frac{L\lambda_{t+1}^2}{2}\left[\hat{F}_{t+1}(\hat{q}_{t+1}) - (1-\alpha)\right]^2$$

$$= -\lambda_{t+1}\left[\hat{F}_{t+1}(\hat{q}_{t+1}) - (1-\alpha)\right]^2 \left[\frac{F(\hat{q}_{t+1}) - (1-\alpha)}{\hat{F}_{t+1}(\hat{q}_{t+1}) - (1-\alpha)} - \frac{L\lambda_{t+1}}{2}\right]$$

To satisfy $\mathbb{E}\left[\ell_{1-\alpha}(s_{t+1} - q_{t+1})|\mathcal{S}_t\right] \leq \mathbb{E}\left[\ell_{1-\alpha}(s_{t+1} - \hat{q}_{t+1})|\mathcal{S}_t\right]$, it suffices that $\lambda_{t+1}$ satisfies:

$$\lambda_{t+1} < \frac{2(F_{t+1}(\hat{q}_{t+1}) - (1-\alpha))}{L(\hat{F}_{t+1}(\hat{q}_{t+1}) - (1-\alpha))},$$

which can be achieved since $\hat{F}_{t+1}(\hat{q}_{t+1}) - (1-\alpha)$ and $F_{t+1}(\hat{q}_{t+1}) - (1-\alpha)$ have the same sign.

Denote $\epsilon_{t+1} = \sup_q |F_{t+1}(q) - \hat{F}_{t+1}(q)|, \delta_{t+1} = |\hat{F}_{t+1}(\hat{q}_{t+1}) - (1-\alpha)| \leq 1 - \alpha$, then

$$\mathcal{L}(q_{t+1}) - \mathcal{L}(\hat{q}_{t+1}) \leq \nabla \mathcal{L}(\hat{q}_{t+1})(q_{t+1} - \hat{q}_{t+1}) + \frac{L}{2}\|q_{t+1} - \hat{q}_{t+1}\|_2^2$$

$$\leq \lambda_{t+1}(\delta_{t+1})(\delta_{t+1} + \epsilon_{t+1}) + \frac{L}{2}\lambda_{t+1}\delta_{t+1}^2$$

$$< \lambda_{t+1}(1 + \epsilon_{t+1} + \frac{L}{2}).$$

Hence, even if the same-sign assumption in Proposition 1 does not hold, the "instantaneous" performance of the refinement step is bounded by $\lambda_{t+1}$ and $\epsilon_{t+1}$. $\qquad \square$

## B.2 REGRET GUARANTEE WITH CONSTANT LEARNING RATES

**Theorem 1.** Let $\ell_t(q) = \ell_{1-\alpha}(s_t - q)$. For arbitrary $\{u_t\}_{t\geq 1}$, $u_0 = 0$, we have:

$$\underbrace{\frac{1}{T}\sum_{t=1}^{T}[\ell_t(q_t) - \ell_t(u_t)]}_{regret} + \underbrace{\frac{\eta(1-2\alpha)}{4}\left(\frac{\sum_{t=1}^{T} err_t}{T} - \alpha\right)}_{coverage} \leq \frac{\eta}{T}\sum_{t=1}^{T}\|\alpha - err_t - M_t\|_2^2$$

$$+ \underbrace{\sum_{t=1}^{T}\frac{1}{2\eta}\left(\|u_t - \hat{q}_t\|_2^2 - \|u_{t-1} - \hat{q}_t\|_2^2\right)}_{environments}.$$

*Proof.* We begin by presenting the Bregman Proximal Inequality (Chen & Teboulle, 1993):

**Lemma 1.** *(Bregman Proximal Inequality): Let $\mathcal{X}$ be a convex set in a Banach space, and $f : \mathcal{X} \to \mathbb{R}$ be a closed proper convex function. Given a convex regularizer $\psi : \mathcal{X} \to \mathbb{R}$, denote its Bregman divergence by $D_\psi(\cdot, \cdot)$. Then, $q_t$ under the update:*

$$q_t = \arg\min_{q \in \mathcal{X}} f(q) + D_\psi(q, q_{t-1})$$

*satisfies for any $u \in \mathbb{R}$,*

$$f(q_t) - f(u) \le D_\psi(u, q_{t-1}) - D_\psi(u, q_t) - D_\psi(q_t, q_{t-1}).$$

In our case, we take $\psi$ to be $\psi(\boldsymbol{x}) = \|\boldsymbol{x}\|_2^2/2$, then $D_\psi(\boldsymbol{x}, \boldsymbol{y}) = \|\boldsymbol{x} - \boldsymbol{y}\|_2^2/2$. By convexity of $\ell_t$, we upper bound the instantaneous dynamic regret to sum of three terms:

$$\ell_t(q_t) - \ell_t(u_t) \le \langle \nabla \ell_t(q_t), q_t - u_t \rangle$$
$$= \underbrace{\langle \nabla \ell_t(q_t) - M_t, q_t - \hat{q}_{t+1} \rangle}_{(a)} + \underbrace{\langle M_t, q_t - \hat{q}_{t+1} \rangle}_{(b)} + \underbrace{\langle \nabla \ell_t(q_t), \hat{q}_{t+1} - u_t \rangle}_{(c)}.$$

For term (a), note that

$$q_t - \hat{q}_{t+1} = \hat{q}_t - \eta M_t - \hat{q}_{t+1} = \eta(\alpha - err_t - M_t),$$

hence

$$(a) = \langle \nabla \ell_t(q_t) - M_t, q_t - \hat{q}_{t+1} \rangle = \eta\|\alpha - err_t - M_t\|_2^2.$$

For term (b) :

$$(b) = \langle M_t, \hat{q}_t - \hat{q}_{t+1} \rangle = \frac{\eta}{2}(\|err_t - \alpha\|_2^2 - \|\alpha - err_t - M_t\|_2^2 - M_t^2)$$
$$\le \frac{\eta}{2}\|err_t - \alpha\|_2^2 - \frac{\eta}{4}\|\alpha - err_t - M_t + M_t\|_2^2$$
$$= \frac{\eta}{4}\|err_t - \alpha\|_2^2.$$

For term (c), using Lemma 1, the update $\hat{q}_{t+1} = \arg\min_q \left\{ \eta\langle \nabla \ell_t(q_t), q \rangle + \frac{1}{2}\|q - q_t\|^2 \right\}$ implies:

$$(c) = \langle \nabla \ell_t(q_t), \hat{q}_{t+1} - u_t \rangle \le \frac{1}{2\eta}\left( \|u_t - \hat{q}_t\|_2^2 - \|u_t - \hat{q}_{t+1}\|_2^2 - \|q_t - \hat{q}_{t+1}\|_2^2 \right)$$
$$= \frac{1}{2\eta}\left( \|u_t - \hat{q}_t\|_2^2 - \|u_t - \hat{q}_{t+1}\|_2^2 \right) - \frac{\eta}{2}\|err_t - \alpha\|_2^2$$

Combine the three upper bounds:

$$\ell_t(q_t) - \ell_t(u_t) \le \eta\|\alpha - err_t - M_t\|_2^2 - \frac{\eta}{4}\|err_t - \alpha\|_2^2 + \frac{1}{2\eta}\left( \|u_t - \hat{q}_t\|_2^2 - \|u_t - \hat{q}_{t+1}\|_2^2 \right)$$
$$\le \eta\|\alpha - err_t - M_t\|_2^2 - \frac{\eta}{4}(1 - 2\alpha)(err_t - \alpha) - \frac{\eta}{4}(\alpha - \alpha^2)$$
$$+ \frac{1}{2\eta}\left( \|u_t - \hat{q}_t\|_2^2 - \|u_t - \hat{q}_{t+1}\|_2^2 \right)$$

Summing over $t$ from 1 to $T$, we have:

$$\sum_{t=1}^{T} [\ell_t(q_t) - \ell_t(u_t)] + \frac{\eta}{4}\sum_{t=1}^{T}(1 - 2\alpha)(err_t - \alpha) + \frac{\eta T}{4}\sum_{t=1}^{T}(\alpha - \alpha^2)$$
$$\le \eta\sum_{t=1}^{T}\|\alpha - err_t - M_t\|_2^2 + \sum_{t=1}^{T}\frac{1}{2\eta}\left( \|u_t - \hat{q}_t\|_2^2 - \|u_t - \hat{q}_{t+1}\|_2^2 \right)$$
$$\le \eta\sum_{t=1}^{T}\|\alpha - err_t - M_t\|_2^2 + \sum_{t=1}^{T}\frac{1}{2\eta}\left( \|u_t - \hat{q}_t\|_2^2 - \|u_{t-1} - \hat{q}_t\|_2^2 \right). \ (u_0 = 0)$$

Dividing both sides by $T$ and ignoring the constant $\frac{\eta}{4}(\alpha - \alpha^2)$ completes the proof. $\qquad\square$

## B.3 REGRET GUARANTEE WITH ARBITRARY LEARNING RATES

To ensure that the regret guarantees with arbitrary learning rates match the form of Theorem 1 and are presented more transparently, we consider the original OOGD:

$$\hat{q}_{t+1} = \hat{q}_t + \eta_t(err_t - \alpha)$$
$$q_{t+1} = \hat{q}_{t+1} - \eta_{t+1}M_{t+1}.$$

Note that in our algorithm, the learning rate used for the second update is $\eta_t$ instead of $\eta_{t+1}$. For arbitray learning rate $\{\eta_t\}_{t\geq 1}$, we have:

$$\frac{1}{T}\sum_{t=1}^{T}[\ell_t(q_t) - \ell_t(u_t)] + C_T\left[(1-2\alpha)\left(\frac{\sum_{t=1}^{T}err_t}{T} - \alpha\right) + \alpha - \alpha^2\right]^2$$
$$\leq \frac{1}{T}\sum_{t=1}^{T}\eta_t\|\alpha - err_t - M_t\|_2^2 + \sum_{t=1}^{T}\frac{1}{2\eta_t}\left(\|u_t - \hat{q}_t\|_2^2 - \|u_t - \hat{q}_{t+1}\|_2^2\right),$$

where $C_T$ is the harmonic mean of the learning rates, i.e.

$$C_T = \frac{T}{\sum_{t=1}^{T}\frac{1}{\eta_t}}.$$

*Proof.* Similar to the proof of Theorem 1, we obtain:

$$\ell_t(q_t) - \ell_t(u_t) \leq \langle\nabla\ell_t(q_t), q_t - u_t\rangle$$
$$= \underbrace{\langle\nabla\ell_t(q_t) - M_t, q_t - \hat{q}_{t+1}\rangle}_{(a)} + \underbrace{\langle M_t, q_t - \hat{q}_{t+1}\rangle}_{(b)} + \underbrace{\langle\nabla\ell_t(q_t), \hat{q}_{t+1} - u_t\rangle}_{(c)},$$

and:

$$(a) = \langle\nabla\ell_t(q_t) - M_t, q_t - \hat{q}_{t+1}\rangle = \eta_t\|\alpha - err_t - M_t\|_2^2.$$

$$(b) \leq \frac{\eta_t}{4}\|err_t - \alpha\|_2^2.$$

$$(c) \leq \frac{1}{2\eta_t}\left(\|u_t - \hat{q}_t\|_2^2 - \|u_t - \hat{q}_{t+1}\|_2^2\right) - \frac{\eta_t}{2}\|err_t - \alpha\|_2^2$$

Combine the three upper bounds:

$$\ell_t(q_t) - \ell_t(u_t) \leq \eta_t\|\alpha - err_t - M_t\|_2^2 - \frac{\eta_t}{4}\|err_t - \alpha\|_2^2 + \frac{1}{2\eta_t}\left(\|u_t - \hat{q}_t\|_2^2 - \|u_t - \hat{q}_{t+1}\|_2^2\right)$$

Summing over $t$ from 1 to $T$, we have:

$$\sum_{t=1}^{T}[\ell_t(q_t) - \ell_t(u_t)] \leq \sum_{t=1}^{T}\eta_t\|\alpha - err_t - M_t\|_2^2 - \sum_{t=1}^{T}\eta_t(err_t - \alpha)^2 + \sum_{t=1}^{T}\frac{1}{2\eta_t}\left(\|u_t - \hat{q}_t\|_2^2 - \|u_t - \hat{q}_{t+1}\|_2^2\right)$$
$$\leq \sum_{t=1}^{T}\eta_t\|\alpha - err_t - M_t\|_2^2 - TC_T\left(\frac{\sum_{t=1}^{T}|err_t - \alpha|^2}{T}\right)^2$$
$$+ \sum_{t=1}^{T}\frac{1}{2\eta_t}\left(\|u_t - \hat{q}_t\|_2^2 - \|u_t - \hat{q}_{t+1}\|_2^2\right) \qquad \text{(Cauchy-Schwarz inequality)}$$
$$= \sum_{t=1}^{T}\eta_t\|\alpha - err_t - M_t\|_2^2 - TC_T\left[(1-2\alpha)\left(\frac{\sum_{t=1}^{T}err_t}{T} - \alpha\right) + \alpha - \alpha^2\right]^2$$
$$+ \sum_{t=1}^{T}\frac{1}{2\eta_t}\left(\|u_t - \hat{q}_t\|_2^2 - \|u_t - \hat{q}_{t+1}\|_2^2\right),$$

which completes the proof. $\qquad\square$

### B.4   PROOFS OF COVERAGE GUARANTEES

Proposition 2 is simply a special case of Theorem 2, so we only prove the more general result of Theorem 2. We first prove the lemma below that shows the boundedness of $q_t$. Lemma 2 is essential in the proof of Theorem 2.

**Lemma 2.** *Fix an initial threshold $q_1 \in [0, B]$. Then COP in equation 1 with arbitrary nonnegative learning rate $\eta_t$ satisfies that*

$$-\Omega_t(2M + 1) \le q_t \le B + \Omega_t(2M + 1) \quad \forall t \ge 1,$$

*where $\Omega_0 = 0$, and $\Omega_t = \max_{1 \le r \le t} \eta_r$ for $t \ge 1$.*

*Proof.* We first prove the upper bound. Combine eq. (4) and eq. (5) we get:

$$q_t = q_{t-1} + \eta_{t-1}(err_{t-1} - \alpha) + \eta_{t-1}M_t - \eta_t M_{t+1}, \tag{15}$$

where $M_t$ is the optimistic term defined in eq. (8). For any $t$, if $q_t < s_t$, we have $q_t < B < B + \Omega_t(2M + 1)$. If $q_t > s_t$, denote $l$ as the largest integer below $t$ satisfying $q_l \le s_l$, then $q_r > s_r$, for $l < r \le t$. Hence,

$$q_r = q_{r-1} - \eta_{r-1}\alpha + \eta_{r-1}M_r - \eta_r M_{r+1}, \quad l < r \le t.$$

Through iteration we obtain :

$$q_t = q_l + \sum_{r=l}^{t-1} [(err_l - \alpha)\eta_r + (\eta_{r-1}M_r - \eta_r M_{r+1})]$$

$$= q_l + \eta_l - \sum_{r=l}^{t-1} \eta_r \alpha + (\eta_{l-1}M_l - \eta_t M_{t+1})$$

$$\le s_l + \eta_l + (\eta_{l-1}M_l - \eta_t M_{t+1})$$

$$\le B + \Omega_t(2M + 1).$$

For the lower bound, if $q_t > s_t$, we have $q_t > 0 > -\Omega_t(2\lambda M + 1)$. If $q_t \le s_t$, denote $l$ as the largest integer below $t$ satisfying $q_l > s_l$, then $q_r \le s_r$, for $l < r \le t$. Hence,

$$q_t = q_l + \sum_{r=l}^{t-1} [(err_l - \alpha)\eta_r + (\eta_{r-1}M_r - \eta_r M_{r+1})]$$

$$= q_l - \eta_l + \sum_{r=l}^{t-1} \eta_r(1 - \alpha) + (\eta_{l-1}M_l - \eta_t M_{t+1})$$

$$> s_l - \eta_l + (\eta_{l-1}M_l - \eta_t M_{t+1})$$

$$\ge -\Omega_t(2M + 1).$$

$\square$

**Theorem 2.** Assume that for each $t$, $s_t \in [0, B]$ and $M_t \in [-M, M]$, then for all $T \ge 1$, the prediction sets generated by Algorithm 1 satisfies

$$\left| \frac{1}{T} \sum_{t=1}^{T} \text{err}_t - \alpha \right| \le \frac{B + (2 + 6M)\Omega_T}{T} \|\Delta_{1:T}\|_1,$$

where $\|\Delta_{1:T}\|_1 = |\eta_1^{-1}| + \sum_{t=2}^{T} |\eta_t^{-1} - \eta_t^{-1}|, \Omega_T = \max_{1 \le r \le T} \eta_r.$

*Proof.*

$$\left| \frac{1}{T} \sum_{t=1}^{T} (\text{err}_t - \alpha) \right| = \left| \frac{1}{T} \sum_{t=1}^{T} \left( \sum_{r=1}^{t} \Delta_r \right) \cdot \eta_t \left( \text{err}_t - \alpha \right) \right|$$

$$= \left| \frac{1}{T} \sum_{r=1}^{T} \Delta_r \left( \sum_{t=r}^{T} \eta_t \left( \text{err}_t - \alpha \right) \right) \right|$$

$$= \left| \frac{1}{T} \sum_{r=1}^{T} \Delta_r \left( q_{T+1} - q_r + \eta_T M_{T+1} - \eta_r M_{r+1} \right) \right|$$

$$\leq \frac{1}{T} \left| \sum_{r=1}^{T} \Delta_r (q_{T+1} - q_r) \right| + \frac{1}{T} \left| \sum_{r=1}^{T} \Delta_r (\eta_T M_{T+1} - \eta_r M_{r+1}) \right|$$

$$\leq \frac{B + (2 + 4M)\Omega_T}{T} \|\Delta_{1:T}\|_1 + \frac{2M\Omega_T \|\Delta_{1:T}\|_1}{T}$$

$$= \frac{B + (2 + 6M)\Omega_T}{T} \|\Delta_{1:T}\|_1.$$

Specifically, if $\eta_t \equiv \eta$, we have

$$\left| \frac{1}{T} \sum_{t=1}^{T} (\text{err}_t - \alpha) \right| \leq \frac{B + (2 + 6M)\eta}{T\eta}$$

□

**Theorem 3.** Assume the scores $s_t \overset{\text{iid}}{\sim} P$ and has a continuous distribution function $F$. The learning rates $\{\eta_t\}$ satisfy:

$$\sum_{t=1}^{\infty} \eta_t = \infty, \ \sum_{t=1}^{\infty} \eta_t^2 < \infty.$$

Let $q^*$ be the $(1-\alpha)$-th quantile of $P$, satisfying: for $q > q^*$, $F(q) > 1 - \alpha$ and for $q < q^*$, $F(q) < 1 - \alpha$, Then $q_t \to q^*$, i.e. $\lim_{t\to\infty} P(Y_t \in C_t(X_t)) = 1 - \alpha$.

*Proof.* Denote random variable $\epsilon_t = err_t - \mathbb{E}err_t = err_t - 1 + F(q_t), S_t = \sum_{i=1}^{t} \eta_i \epsilon_i, A_t = \sum_{i=t}^{\infty} \eta_i \epsilon_i, \mathcal{F}_t = \sigma(s_1, \cdots, s_t)$. We first prove $A_t \xrightarrow{\text{a.s.}} 0$.

Note that $\mathbb{E}(\eta_{t+1}\epsilon_{t+1}|\mathcal{F}_t) = 0$ and for $j > i$, $\mathbb{E}(\epsilon_i \epsilon_j) = \mathbb{E}\left[\mathbb{E}\epsilon_i \epsilon_j | \mathcal{F}_{j-1}\right] = \mathbb{E}\left[\epsilon_i (\mathbb{E}\epsilon_j | \mathcal{F}_{j-1})\right] = 0$. For each $t \geq 1$, $\mathbb{E}(S_{t+1}|\mathcal{F}_t) = S_t + \mathbb{E}(\eta_{t+1}\epsilon_{t+1}|\mathcal{F}_t) = S_t$. Hence, $\{S_t\}_{t\geq 1}$ is a martingale with respect to the filtration $\mathcal{F}_t$. Further,

$$(\mathbb{E}|S_t|)^2 \leq \mathbb{E}|S_t|^2 = \sum_{i=1}^{t} \eta_i^2 \mathbb{E}\epsilon_i^2 \leq \sum_{i=1}^{t} \eta_i^2 < \infty.$$

Applying Doob's first martingale convergence theorem, we obtain that $\{S_t\}_{t\geq 1}$ converges almost surely. Therefore, $A_t \xrightarrow{\text{a.s.}} 0$.

Next, let $g(x) = 1 - \alpha - F(x), p_t = q_t + \eta_{t-1}M_t + A_t$. We have:

$$p_{t+1} - p_t = q_{t+1} - q_t + \eta_t M_{t+1} - \eta_{t-1}M_t + A_{t+1} - A_t$$
$$= \eta_t(err_t - \alpha) - \eta_t(err_t - 1 + F(q_t)) \quad \text{(by 15)}$$
$$= \eta_t(1 - \alpha - F(q_t))$$
$$= \eta_t g(p_t - A_t - \eta_{t-1}M_t).$$

By Lemma 2, $q_t$ is bounded, hence $\{p_t\}$ is bounded, a.s.. By Bolzano–Weierstrass theorem, it has a convergent subsequence $\{p_{u_t}\}$. We now prove $\lim_{u_t \to \infty} p_{u_t} = 0$ by contradiction.

Suppose $\lim_{t \to \infty} p_{u_t} = q^* + 3\delta, \delta > 0$. The case of $\lim_{t \to \infty} p_{u_t} < 0$ follows by the same argument. Since $A_t + \eta_{t-1} M_t \xrightarrow{a.s.} 0$, there exists $t_0 \in \mathbb{N}$ s.t. $\forall t \geq t_0$, $|A_t + \eta_{t-1} M_t| < \delta$, $\eta_t < \delta$, $p_{u_t} - q^* > 2\delta$. For $t > t_0$,

$$p_{u_t - 1} = p_{u_t} - \eta_{u_t - 1} g(p_{u_t - 1} - A_{u_t - 1} - \eta_{u_t - 2} M_{u_t - 1}) \geq p_{u_t} - \eta_{u_t - 1} > q^* + \delta,$$

hence

$$p_{u_t} = p_{u_t - 1} + \eta_{u_t - 1} g(p_{u_t - 1} - A_{u_t - 1} - \eta_{u_t - 2} M_{u_t - 1})$$
$$\leq p_{u_t - 1} + \eta_{u_t - 1} g(p_{u_t - 1} - \delta) \leq p_{u_t - 1} + \eta_{u_t - 1} g(q^*) = p_{u_t - 1},$$

i.e. $p_{u_t - 1} \geq p_{u_t}$. By induction we can prove that for any $t > t_0$, $p_{u_{t-1}} \geq p_{u_{t-1}+1} \geq \cdots \geq p_{u_t} > q^* + 2\delta$.

Next, let $V(p) = (p - q^*)^2$, then for any $t > u_{t_0}$, we have:

$$
\begin{aligned}
V(p_{t+1}) - V(p_t) &= (p_t + \eta_t g(p_t - A_t - \eta_{t-1} M_t) - q^*)^2 - (p_t - q^*)^2 \\
&= 2\eta_t (p_t - q^*) g(p_t - A_t - \eta_{t-1} M_t) + \eta_t^2 [g(p_t - A_t - \eta_{t-1} M_t)]^2 \\
&\leq 2\eta_t \cdot 2\delta g(q^* + \delta) + \eta_t^2 \\
&= -4\eta_t \delta(F(q^* + \delta) - F(q^*)) + \eta_t^2.
\end{aligned}
\tag{1}
$$

By assumption, $F(q^* + \delta) - F(q^*) > 0$. As $\sum_{t=u_{t_0}} \eta_t = \infty$ and $\sum_{t=u_{t_0}} \eta_t^2 < \infty$, summing equation 1 from $u_{t_0}$ to $\infty$ gives that $V_t \to -\infty$, contradicting the convergence of $\{p_{u_t}\}_{t \geq 1}$.

Finally, we conclude that any convergent subsequence of $\{p_t\}_{t \geq 1}$ converges to $q^*$. Therefore, $\lim_{t \to \infty} p_t = q^*$. Using $A_t \to 0$ and $\eta_{t-1} M_t \to 0, t \to \infty$ completes the proof. $\qquad\square$

## C  IMPLEMENT OF COP WITH KERNEL-BASED CDF

In this section, we compare the performance of using ECDF and Gaussian kernel density estimator (KDE) for estimating the CDF in the COP framework $\hat{F}_{t+1}$. The KDE is implemented using a sliding window approach to incorporate recent data points and adapt to distribution shifts. We set the scale factor $\lambda = 0.5$, the length window $w = 100$, and the bandwidth $h$ of Silverman's rule:

$$h = 0.9 \times \sigma \times w^{-0.2},$$

where $\sigma$ is the minimum of the standard deviation of the window data and the interquartile range (IQR) divided by $1.34$. Then the CDF value at $q$ is estimated using the Gaussian kernel:

$$\hat{F}_{t+1}(\hat{q}_{t+1}) = \frac{1}{n} \sum_{i=t-w+1}^{t} \Phi(\frac{\hat{q}_{t+1} - s_i}{h}),$$

where $\Phi$ is the cumulative distribution function of the standard normal distribution, $s_i$ is the non-conformity score at timestep $t$.

We evaluate the performance of the choices of estimated CDF, including empirical CDF (denoted ECDF) and kernel-based CDF (denoted Kernel), across six datasets: Changepoint, Distribution Drift, Amazon Stock, Google Stock, Electricity Demand, and Temperature. The results are presented in Table 3. We can see that even with simple ECDF, COP can achieve tight prediction sets and maintain coverage rates close to the nominal level. The kernel-based CDF shows comparable performance in some cases, but may require further tuning of parameters such as bandwidth for optimal results.

## D  LEARNING RATE SELECTION IN THE EXPERIMENTS

In our experiments, the default setting is to select four learning rates on all datasets and then pick the one that yields the best performance. Considering the high sensitivity of LQT and OGD-based methods to the learning rate, we carefully select more than four candidate learning rates for these methods across various datasets. The following is the list of all learning rates used, which are

Table 3: The experimental results on the choices of estimated CDF with nominal level $\alpha = 10\%$.

| Dataset | Method | Prophet | | | AR | | | Theta | | |
|---|---|---|---|---|---|---|---|---|---|---|
| | | Coverage ( %) | Average width | Median width | Coverage ( %) | Average width | Median width | Coverage ( %) | Average width | Median width |
| Changepoint | ECDF | 89.8 | 8.29 | 8.44 | 89.7 | 8.18 | 8.25 | 89.8 | 8.45 | 8.53 |
| | Kernel | 89.8 | 8.29 | 8.45 | 89.7 | 8.11 | 8.21 | 89.8 | 8.45 | 8.53 |
| Distribution | ECDF | 90.6 | 7.07 | 6.89 | 90.0 | 7.09 | 6.97 | 90.9 | 7.30 | 6.99 |
| Drift | Kernel | 90.6 | 7.07 | 6.89 | 90.0 | 7.10 | 6.98 | 90.9 | 7.30 | 6.99 |
| Amazon | ECDF | 89.6 | 39.86 | 27.91 | 89.5 | 17.09 | 12.90 | 89.6 | 17.21 | 12.23 |
| Stock | Kernel | 89.6 | 40.32 | 27.98 | 89.4 | 17.23 | 12.95 | 89.6 | 17.35 | 12.56 |
| Google | ECDF | 89.7 | 49.72 | 42.09 | 89.6 | 19.87 | 17.04 | 89.3 | 30.25 | 28.24 |
| Stock | Kernel | 89.7 | 50.85 | 42.33 | 89.6 | 19.92 | 16.94 | 89.3 | 30.36 | 28.40 |
| Electricity | ECDF | 90.1 | 0.385 | 0.376 | 90.0 | 0.117 | 0.098 | 89.8 | 0.069 | 0.052 |
| Demand | Kernel | 90.1 | 0.384 | 0.377 | 90.0 | 0.118 | 0.099 | 89.8 | 0.069 | 0.052 |
| Temperature | ECDF | 90.1 | 7.05 | 7.07 | 89.9 | 5.85 | 5.58 | 90.0 | 6.27 | 6.18 |
| | Kernel | 90.1 | 7.06 | 7.08 | 89.9 | 5.85 | 5.59 | 90.0 | 6.25 | 6.17 |

determined through this systematic selection process to ensure fair and effective comparison.

$$\text{ACI} : \eta = \{0.1, 0, 05, 0.01, 0.005\},$$
$$\text{OGD} : \eta = \{10, 5, 1, 0.5, 0.1, 0.05, 0.01, 0.005\},$$
$$\text{SF-OGD} : \eta = \{1000, 500, 100, 50, 10, 5, 1, 0.5, 0.1, 0.05\},$$
$$\text{decay-OGD} : \eta = \{2000, 1000, 200, 100, 20, 10, 2, 1, 0.2, 0.1\},$$
$$\text{Conformal PID} : \eta = \{1, 0.5, 0.1, 0.05\},$$
$$\text{ECI} : \eta = \{1, 0.5, 0.1, 0.05\},$$
$$\text{LQT} : \eta = \{10, 5, 1, 0.5, 0.1, 0.05, 0.01\},$$
$$\text{COP} : \eta = \{1, 0.5, 0.1, 0.05\}$$

For ACI and OGD, they do not use adaptive learning rates. For SF-OGD:

$$\eta_t = \eta \cdot \frac{\nabla \ell^{(t)}(q_t)}{\sqrt{\sum_{i=1}^{t} \|\nabla \ell^{(i)}(q_i)\|_2^2}},$$

where $\ell^{(t)}(q_t)$ is quantile loss and $q_t$ is the predicted radius at time $t$. For decay-OGD:

$$\eta_t = \eta \cdot t^{-\frac{1}{2} - \epsilon},$$

where the hyperparameter $\epsilon = 0.1$ follows Angelopoulos et al. (2024). For conformal PID, ECI and COP:

$$\eta_t = \eta \cdot (\max\{s_{t-w+1}, \cdots, s_t\} - \min\{s_{t-w+1}, \cdots, s_t\}),$$

where $s_t$ is the non-conformity score at time $t$ and the window length $w = 100$ follows Angelopoulos et al. (2023b).

## E   MORE EXPERIMENTAL RESULTS IN SIMULATION DATASETS

We also evaluated our method in three other simulation datasets under variance changepoint, heavy-tailed noise, and extreme distribution drift setting, respectively. Both datasets $\{X_i, Y_i\}_{i=1}^{n}$ are generated according to a linear model $Y_t = X_t^T \beta_t + \epsilon_t$, $X_t \sim \mathcal{N}(0, I_4)$, $\epsilon_t \sim \mathcal{N}(0, \sigma_t)$, $n = 2000$.

- Variance changepoint setting: we set fixed $\beta = (2, 1, 0.5, -0.5)^\top$ and two variance change-points: $\sigma_t = 1$ for $t = 1, \ldots, 500$; $\sigma_t = 3$ for $t = 501, \ldots, 1500$; and $\sigma_t = 0.5$ for $t = 1501, \ldots, 2000$.

- Heteroskedastic and heavy-tailed noise setting: we set fixed $\beta = (2, 1, 0.5, -0.5)^\top$ and the noise $\epsilon_t$ is $t(2)$, with standard deviation $1 + 2|X_t^T \beta|^3 / \mathbb{E}(|X^T \beta|^3)$.

- Extreme distribution drift setting: we set $\beta_1 = (20, 10, 1, 1)^\top$, $\beta_n = (1, 1, 20, 10)^\top$, and use linear interpolation to compute $\beta_t = \beta_1 + \frac{t-1}{n-1}(\beta_n - \beta_1)$.

Table 4: The experimental results in the three other simulation datasets with nominal level $\alpha = 10\%$. Note that, since PID, ECI and LQT methods completely fail on the Extreme Drift dataset, we did not include them in the table.

| Dataset | Method | Prophet | | | AR | | | Theta | | |
|---|---|---|---|---|---|---|---|---|---|---|
| | | Coverage (%) | Average width | Median width | Coverage (%) | Average width | Median width | Coverage (%) | Average width | Median width |
| Variance Changepoint | ACI | 91.0 | $\infty$ | 11.06 | 91.0 | $\infty$ | 10.74 | 91.0 | $\infty$ | 10.90 |
| | OGD | 90.1 | 10.57 | 10.65 | 90.0 | 10.43 | 10.25 | 90.0 | 10.44 | 10.07 |
| | SF-OGD | 90.0 | 14.75 | 14.06 | 90.0 | 14.80 | 14.18 | 90.0 | 14.47 | 13.82 |
| | decay-OGD | 90.2 | 10.71 | 11.21 | 89.9 | 10.41 | 11.00 | 90.2 | 10.76 | 11.12 |
| | PID | 89.7 | 13.17 | 11.81 | 89.7 | 13.07 | 11.77 | 89.7 | 13.26 | 11.99 |
| | ECI | 89.9 | 10.66 | 10.39 | 89.8 | 10.35 | 9.75 | 89.9 | 10.60 | 10.04 |
| | LQT | 90.6 | 12.10 | 11.77 | 88.7 | 10.84 | 10.97 | 89.8 | 11.64 | 11.38 |
| | COP | 89.9 | 10.78 | 10.79 | 89.8 | 10.37 | 9.87 | 89.9 | 10.66 | 10.10 |
| Heavy-tailed | ACI | 90.3 | $\infty$ | 10.03 | 90.2 | $\infty$ | 10.01 | 90.3 | $\infty$ | 9.82 |
| | OGD | 90.0 | 9.94 | 9.95 | 90.0 | 9.91 | 9.95 | 90.2 | 10.18 | 10.25 |
| | SF-OGD | 90.0 | 15.22 | 14.05 | 90.0 | 15.10 | 13.93 | 90.0 | 15.43 | 13.91 |
| | decay-OGD | 90.3 | 9.97 | 10.01 | 89.9 | 9.69 | 9.77 | 90.8 | 10.10 | 9.94 |
| | PID | 89.7 | 13.60 | 11.47 | 89.6 | 13.14 | 11.36 | 89.6 | 13.12 | 11.19 |
| | ECI | 89.9 | 10.54 | 10.49 | 90.0 | 10.34 | 10.27 | 90.0 | 10.55 | 10.36 |
| | LQT | 92.0 | 13.21 | 12.32 | 89.4 | 10.13 | 10.09 | 92.8 | 15.79 | 13.74 |
| | COP | 89.8 | 9.56 | 9.79 | 89.9 | 10.38 | 10.27 | 90.4 | 9.87 | 10.03 |
| Extreme Drfit | ACI | 90.4 | $\infty$ | 79.01 | 89.7 | $\infty$ | 61.85 | 90.3 | $\infty$ | 70.06 |
| | OGD | 91.1 | 275.87 | 280.00 | 89.9 | 64.03 | 62.00 | 91.3 | 213.10 | 212.00 |
| | SF-OGD | 92.0 | 423.68 | 445.98 | 89.9 | 68.23 | 63.62 | 92.4 | 376.03 | 388.03 |
| | decay-OGD | 89.7 | 265.81 | 274.93 | 90.0 | 64.50 | 60.32 | 91.6 | 246.51 | 248.24 |
| | COP | 91.1 | 275.82 | 279.84 | 89.9 | 64.10 | 62.18 | 91.3 | 213.54 | 211.91 |

The results are shown in Table 4. We observe that COP consistently maintains the coverage rate close to the nominal level of $90\%$ in all scenarios while producing competitive interval widths. Notably, in the Variance Changepoint and Heavy-tailed settings, COP achieves average widths comparable to or tighter than OGD and ECI, without the instability seen in ACI (which yields infinite widths). In the Extreme Drift setting, traditional methods like PID, ECI, and LQT failed to construct valid prediction sets due to the severity of the shift, and are thus excluded from the table. In contrast, COP successfully adapts to the extreme drift, maintaining valid coverage similar to OGD but with slightly tighter intervals in the Theta base predictor case. These results further validate the robustness of COP in handling complex distributional shifts.

## F  POST-SHIFT COVERAGE RECOVERY TIME

To quantify the responsiveness of COP and competing baselines after an abrupt distribution shift, we define post-shift recovery time based on the stability of short-horizon empirical coverage. Let $t_c$ denote the changepoint index. For each method, we compute a sliding-window coverage rate

$$cvg_r(t) = \frac{1}{w_r} \sum_{i=t-w_r+1}^{t} \mathbb{1}\{s_i < q_i\},$$

where $w_r = 20$ is the coverage-estimation window (independent of any internal calibration window). Recovery is declared at the earliest time $t_r > t_c$ such that

$$1 - \alpha - 1/w_r \leq cvg_r(t) \leq 1 - \alpha + 1/w_r \text{ for } k \text{ consecutive indices } t = t_r, \ldots, t_{r+k-1},$$

with nominal coverage $1 - \alpha = 0.9$ and $k = 10$. This criterion requires the local empirical coverage to remain around the target level for multiple consecutive checks, preventing spurious early detections due to stochastic noise.

Table 5: Post-shift coverage recovery time (in steps) in the Changepoint datasets.

| Method | Prophet | | AR | | Theta | |
|---|---|---|---|---|---|---|
| | Recovery Time 1 | Recovery Time 2 | Recovery Time 1 | Recovery Time 2 | Recovery Time 1 | Recovery Time 2 |
| ACI | 35 | 73 | 50 | 0 | 35 | 66 |
| OGD | 12 | 0 | 40 | 0 | 40 | 0 |
| SF-OGD | 12 | 0 | 12 | 0 | 12 | 0 |
| decay-OGD | 6 | 73 | 153 | 73 | 60 | 73 |
| PID | 40 | 0 | 12 | 0 | 40 | 0 |
| ECI | 12 | 8 | 40 | 0 | 40 | 0 |
| LQT | 156 | 73 | 60 | 73 | 60 | 73 |
| COP | 12 | 0 | 40 | 0 | 40 | 0 |

## G  ROBUSTNESS OF INACCURATE ESTIMATED CDF

To evaluate the robustness of COP in adversarial environments where the estimated CDF may be inaccurate, we performed an additional experiment in which the ECDF is corrupted by randomized noise. Specifically, at each time step, we construct a noisy ECDF by

$$\hat{F}_{noisy} = \gamma \hat{F} + (1 - \gamma)\epsilon,$$

where $\hat{F}$ is the ECDF, $\epsilon \sim U(0,1)$ denotes a uniform random noise, and $\gamma \in \{1, 0.9, 0.5, 0.1, 0\}$ controls the level of the corruption. The case $\gamma = 1$ corresponds to the true ECDF, while $\gamma = 0$ represents a purely adversarial environment in which the estimated CDF contains no information about the data distribution. We apply this noise independently at each time step and use $\hat{F}_{noisy}$ in place of $\hat{F}$ in the COP update rule.

Table 6: The experimental results under different noise level $\gamma$ in the two real-world datasets with nominal level $\alpha = 10\%$.

| Dataset | $\gamma$ | Prophet | | | AR | | | Theta | | |
|---|---|---|---|---|---|---|---|---|---|---|
| | | Coverage (%) | Average width | Median width | Coverage (%) | Average width | Median width | Coverage (%) | Average width | Median width |
| Amazon Stock | 1.0 | 89.6 | 39.86 | 27.91 | 89.5 | 17.09 | 12.90 | 89.6 | 17.21 | 12.23 |
| | 0.9 | 89.6 | 38.95 | 27.85 | 89.3 | 17.14 | 12.71 | 89.6 | 17.45 | 13.12 |
| | 0.5 | 89.7 | 42.24 | 29.56 | 89.4 | 17.24 | 13.08 | 89.6 | 17.20 | 12.76 |
| | 0.1 | 90.1 | 46.93 | 30.89 | 89.2 | 17.45 | 12.44 | 89.6 | 17.60 | 12.73 |
| | 0.0 | 90.0 | 62.65 | 53.09 | 89.7 | 22.07 | 18.26 | 89.5 | 30.76 | 28.56 |
| Google Stock | 1.0 | 89.7 | 49.72 | 42.09 | 89.6 | 19.87 | 17.04 | 89.3 | 30.25 | 28.24 |
| | 0.9 | 89.8 | 49.99 | 42.06 | 89.6 | 19.81 | 16.92 | 89.4 | 30.43 | 27.79 |
| | 0.5 | 90.0 | 54.57 | 46.08 | 89.6 | 19.84 | 17.33 | 89.4 | 29.82 | 27.27 |
| | 0.1 | 90.0 | 61.25 | 51.91 | 89.6 | 19.98 | 17.23 | 89.5 | 30.68 | 28.18 |
| | 0.0 | 90.8 | 86.26 | 55.43 | 90.0 | 30.04 | 20.32 | 89.9 | 30.24 | 20.76 |

Table 6 shows the results for Amazon and Google stock datasets across different noisy levels $\gamma$. The coverage rate remains stable across all values of $\gamma$. Even when the CDF is fully replaced by uniform

noise ($\gamma = 0$), the coverage rate varies by at most 0.5%. In contrast, the average and median interval widths exhibit a relatively monotonic trend: as $\gamma$ decreases and the CDF becomes less informative, COP widens its intervals adaptively for validity. This behavior demonstrates that COP can flexibly utilize the CDF information to tighten intervals when the CDF is accurate but to revert to conservative intervals when the CDF is adversarial.

Overall, these results indicate that COP retains near-nominal coverage even when the CDF is corrupted or completely uninformative, thereby confirming its robustness under adversarial noisy CDF.

## H  SENSITIVITY ANALYSIS OF THE SCALE FACTOR

In this section, we will analyze the sensitivity of the scale factor $\lambda$. As defined in the update rule

$$q_{t+1} = \hat{q}_{t+1} - \lambda \left( \hat{F}_{t+1}(\hat{q}_{t+1}) - (1 - \alpha) \right),$$

this parameter governs the magnitude of the refinement step derived from the estimated CDF. We evaluated the performance in the Amazon Stock and Google Stock datasets for three distinct values: $\lambda \in \{0.1, 0.5, 1.0\}$.

Table 7: The experimental results under different scale factor $\lambda$ in the two real-world datasets with nominal level $\alpha = 10\%$.

| Dataset | $\lambda$ | Prophet | | | AR | | | Theta | | |
|---|---|---|---|---|---|---|---|---|---|---|
| | | Coverage ( %) | Average width | Median width | Coverage ( %) | Average width | Median width | Coverage ( %) | Average width | Median width |
| Amazon Stock | 1.0 | 89.4 | 40.31 | 28.96 | 89.0 | 16.97 | 12.88 | 89.5 | 17.01 | 12.2 |
| | 0.5 | 89.6 | 39.86 | 27.91 | 89.5 | 17.09 | 12.9 | 89.6 | 17.21 | 12.23 |
| | 0.1 | 89.7 | 40.98 | 29.06 | 89.3 | 16.90 | 12.62 | 89.6 | 17.26 | 12.66 |
| Google Stock | 1.0 | 89.7 | 52.20 | 42.83 | 89.3 | 19.93 | 16.98 | 89.3 | 30.49 | 28.56 |
| | 0.5 | 89.7 | 49.72 | 42.09 | 89.6 | 19.87 | 17.04 | 89.3 | 30.25 | 28.24 |
| | 0.1 | 89.8 | 52.47 | 43.72 | 89.6 | 19.85 | 17.34 | 89.6 | 30.66 | 28.26 |

The results are shown in Table 7. Overall, COP demonstrates high robustness to variations in the scale factor. Across all base predictors (Prophet, AR, and Theta), the coverage rates remain stable and consistently close to the target, regardless of the specific $\lambda$ chosen. Regarding efficiency, the setting of $\lambda = 0.5$ generally yields the most favorable trade-off, achieving tighter average and median widths compared to the more conservative $\lambda = 0.1$. While $\lambda = 1.0$ also produces competitive widths, it occasionally results in slightly lower coverage rates (e.g., AR on Amazon Stock). These empirical findings justify our default hyperparameter selection of $\lambda = 0.5$ used in the main experiments.

## I  STATISTICAL SIGNIFICANCE ANALYSIS

We regenerated the Changepoint and Distribution Drift datasets using ten different random seeds. For each method and each base model (Prophet, AR, Theta), we evaluated the coverage rate, the average interval width, and the median width on every dataset and then reported the mean and standard deviation across the ten generated datasets.

As shown in Table 8 and Table 9, the variability across multiple data generation seeds is small: coverage rate varies by less than 0.5%, and interval width variations vary slightly. This indicates that performance differences between methods are not attributable to any single random dataset. In both experimental settings, COP consistently achieves coverage rates close to the target level while yielding tight interval widths.

Moreover, we conducted paired t-tests to reflect the statistical significance of shorter width of COP. All of the baselines above show $p$-values less than 0.05 (e.g., OGD has $p$-value=0.014, SF-OGD has $p$-value=0.0002, and ECI's $p$-value=0.03). These results confirm that the observed differences in width between our methods and the COP baseline are statistically significant.

Table 8: The experimental results in the Changepoint datasets with nominal level $\alpha = 10\%$. Values represent the mean ± standard deviation of coverage Rate, average width, and median width across ten independent runs using different random seeds.

| Method | Prophet | | | AR | | | Theta | | |
|---|---|---|---|---|---|---|---|---|---|
| | Coverage | Average | Median | Coverage | Average | Median | Coverage | Average | Median |
| | ( %) | width | width | ( %) | width | width | ( %) | width | width |
| ACI | 90.0±0.01 | $\infty$ | 8.27±0.04 | 90.0±0.01 | $\infty$ | 8.15±0.04 | 90.0±0.01 | $\infty$ | 8.33±0.05 |
| OGD | 90.1±0.02 | 8.63±0.09 | 8.62±0.10 | 90.0±0.02 | 8.46±0.10 | 8.48±0.12 | 90.1±0.03 | 8.68±0.12 | 8.64±0.13 |
| SF-OGD | 90.0±0.01 | 12.65±0.12 | 11.59±0.12 | 90.0±0.01 | 12.66±0.15 | 11.63±0.15 | 90.0±0.01 | 12.68±0.25 | 11.71±0.28 |
| decay-OGD | 90.6±0.10 | 8.52±0.20 | 8.34±0.15 | 90.1±0.09 | 8.17±0.22 | 8.11±0.17 | 90.5±0.30 | 8.43±0.30 | 8.29±0.20 |
| PID | 89.7±0.01 | 11.06±0.15 | 9.36±0.18 | 89.7±0.09 | 10.86±0.10 | 9.18±0.13 | 89.7±0.02 | 10.99±0.20 | 9.32±0.25 |
| ECI | 89.9±0.01 | 8.32±0.17 | 8.34±0.19 | 89.9±0.09 | 8.22±0.23 | 8.26±0.22 | 89.9±0.02 | 8.38±0.17 | 8.43±0.19 |
| LQT | 90.4±0.30 | 10.72±1.20 | 9.70±1.00 | 90.0±0.30 | 10.23±1.00 | 9.40±0.90 | 90.2±0.50 | 10.49±0.90 | 9.70±0.95 |
| COP | 89.8±0.15 | 8.29±0.15 | 8.35±0.15 | 89.4±0.50 | 8.12±0.20 | 8.23±0.20 | 89.8±0.30 | 8.35±0.15 | 8.38±0.18 |

Table 9: The experimental results in the Distribution Drift datasets with nominal level $\alpha = 10\%$. Values represent the mean ± standard deviation of coverage Rate, average width, and median width across ten independent runs using different random seeds.

| Method | Prophet | | | AR | | | Theta | | |
|---|---|---|---|---|---|---|---|---|---|
| | Coverage | Average | Median | Coverage | Average | Median | Coverage | Average | Median |
| | ( %) | width | width | ( %) | width | width | ( %) | width | width |
| ACI | 89.8±0.02 | $\infty$ | 6.61±0.03 | 89.7±0.03 | $\infty$ | 6.53±0.02 | 89.8±0.02 | $\infty$ | 6.64±0.02 |
| OGD | 90.2±0.01 | 7.04±0.03 | 6.91±0.03 | 90.1±0.02 | 6.89±0.04 | 6.84±0.03 | 90.2±0.01 | 7.08±0.04 | 6.98±0.05 |
| SF-OGD | 90.0±0.00 | 11.50±0.01 | 10.35±0.02 | 90.0±0.00 | 11.44±0.06 | 10.28±0.03 | 90.0±0.00 | 11.50±0.08 | 10.28±0.05 |
| decay-OGD | 90.4±0.12 | 7.38±0.10 | 6.76±0.03 | 90.0±0.04 | 6.94±0.06 | 6.60±0.03 | 90.2±0.08 | 7.24±0.05 | 6.74±0.02 |
| PID | 89.7±0.00 | 9.59±0.06 | 7.79±0.02 | 89.6±0.01 | 9.67±0.10 | 7.75±0.02 | 89.7±0.00 | 9.70±0.05 | 7.86±0.01 |
| ECI | 90.0±0.00 | 7.01±0.05 | 6.84±0.03 | 89.9±0.03 | 6.83±0.04 | 6.81±0.04 | 90.1±0.03 | 7.10±0.08 | 6.93±0.04 |
| LQT | 90.7±0.05 | 9.47±0.07 | 8.53±0.03 | 90.5±0.59 | 8.56±0.07 | 8.00±0.20 | 90.5±0.59 | 9.50±0.33 | 8.55±0.04 |
| COP | 90.5±0.11 | 6.77±0.08 | 6.70±0.04 | 89.9±0.03 | 6.67±0.18 | 6.67±0.10 | 90.5±0.14 | 7.12±0.16 | 6.87±0.05 |

Overall, these repeated-data experiments confirm that our findings are not sensitive to the choice of a single random seed. The overall patterns reported in the main text remain the same across independently generated datasets.

## J  COMPUTATIONAL COMPLEXITY ANALYSIS

To compare the practical efficiency of the baselines and COP, we measured the average computation time for each method. All timings were obtained using a single CPU core (Intel(R) Xeon(R) Platinum 8269CY CPU @ 2.50GHz) on the same machine, with each method running for 3020 steps. As shown in Table 10, the reported value represents the mean per-step time cost.

Table 10: Average time cost per update step.

| Method | ACI | OGD | SF-OGD | decay-OGD | PID | ECI | LQT | COP |
|---|---|---|---|---|---|---|---|---|
| Time Cost (ms) | 0.043 | 0.001 | 0.012 | 0.001 | 0.013 | 0.007 | 0.010 | 0.011 |

The simplest gradient-based methods (OGD and decay-OGD) are the fastest, requiring only 0.001 ms per step. Methods that consider all past data, such as SF-OGD and PID, incur higher overhead (around $0.012 - 0.013$ ms). Methods that consider the past data in a certain window, such as ECI, LQT, and COP fall in a similar range ($0.007 - 0.011$ ms). Overall, the differences between methods remain small in absolute terms, and all methods run easily in real time for typical streaming applications.

# K VISUALIZATION OF COVERAGE AND INTERVAL

Figure 1 shows the corresponding rolling-window coverage for each method. The coverage trajectories show how frequently each approach stays close to the target level and how often it experiences deviations. The worse methods tend to display larger swings in coverage, including occasional periods of undercoverage. In contrast, COP and several ECI variants maintain a more stable coverage across the entire time period, even during intervals with sharp changes in the underlying series.

In Figure 2, we plot the interval widths for all methods, together with zoomed-in panels that highlight periods of rapid market movement. These plots show clear differences in how each method responds to changes in volatility. Some approaches, such as PID or ECI-based variants, exhibit sharp jumps or sudden drops in interval width when the underlying series becomes more volatile. Others, including ECI-based methods and COP, adjust their intervals more gradually and maintain a smoother trajectory.

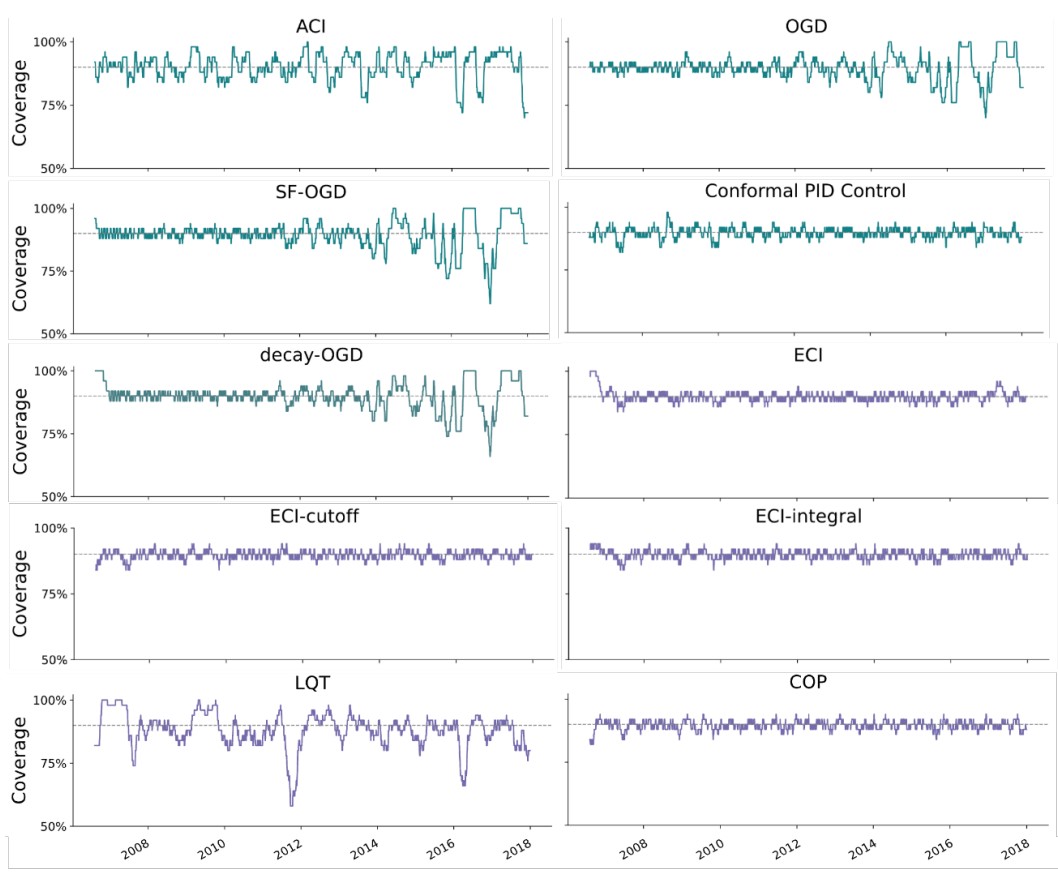

Figure 1: Comparison results of coverage rate on Amazon stock dataset with Prophet model. The coverage is averaged over a rolling window of 50 points.

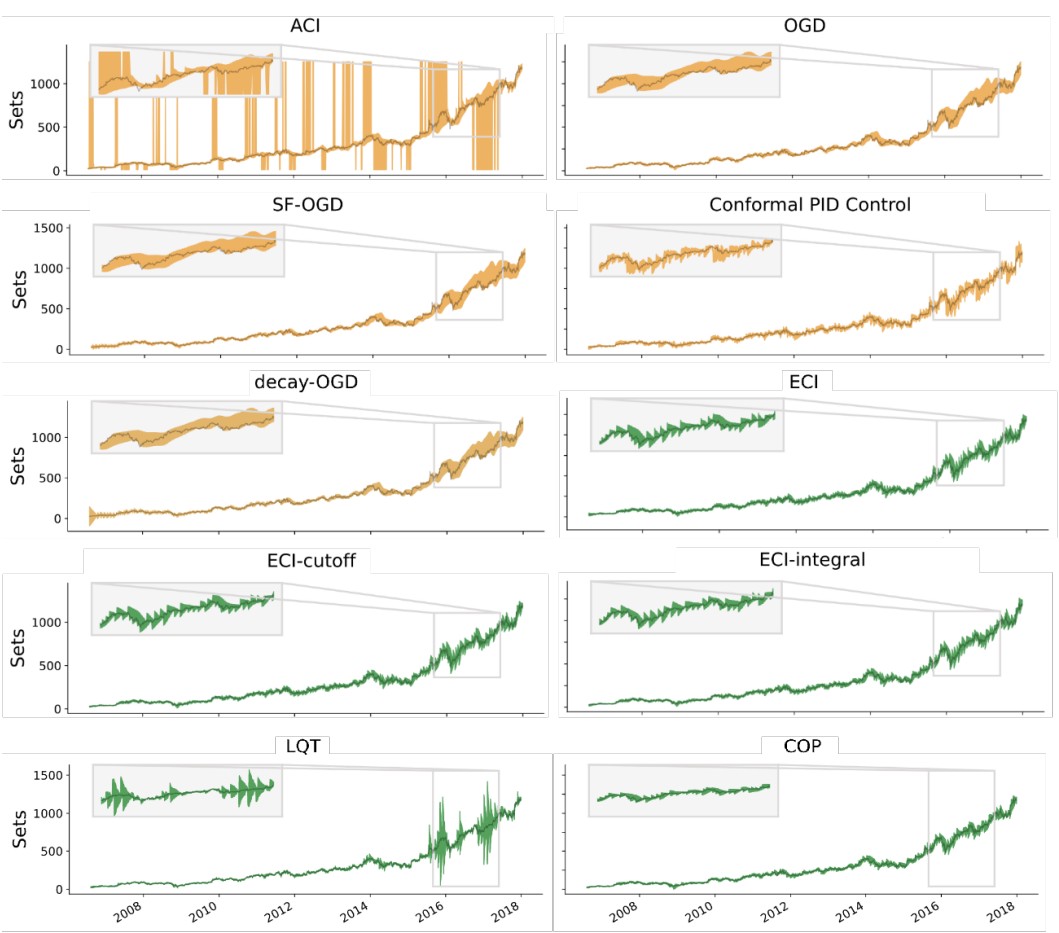

Figure 2: Comparison results of prediction sets on Amazon stock dataset with Prophet model.

