# OpenReview forum: "Distribution-informed Online Conformal Prediction"
_ICLR.cc/2026/Conference — ICLR 2026 Poster_

### Official Review · Reviewer_nxNU · 2025-10-18

**Soundness:** 3
**Presentation:** 1
**Contribution:** 2
**Rating:** 2
**Confidence:** 4

**Summary:**

This paper introduces Conformal Optimistic Prediction (COP), an online conformal prediction algorithm designed to address the over-conservatism of existing methods in non-stationary environments like time series.

The core innovation of COP is a two-step update mechanism that refines the standard gradient-based adjustment for the prediction set size. After a primary update based on the current miscoverage error, COP applies a refinement step using an estimated cumulative distribution function (CDF) of the non-conformity scores. This estimated CDF acts as an "optimistic hint" about the data's underlying patterns, enabling the algorithm to produce tighter, more efficient prediction sets when such patterns exist.

Crucially, the method is robust; it maintains provable, distribution-free, finite-sample coverage guarantees even if the CDF estimate is inaccurate. The authors frame COP within the Optimistic Online Gradient Descent (OOGD) paradigm, deriving a novel joint bound on regret and coverage that clarifies its theoretical motivation.

Experiments on synthetic datasets with distribution shifts and real-world time series from finance, energy, and climate show that COP consistently achieves the target coverage while generating significantly narrower prediction intervals than state-of-the-art baselines, demonstrating its superior efficiency and practical utility.

**Strengths:**

*   **Originality**
    *   Presents a new synthesis of online Conformal Prediction (CP) with Optimistic Online Gradient Descent (OOGD).
    *   Uniquely frames distributional information as a proactive "optimistic hint" rather than simply replacing the robust reactive update rule.

*   **Quality**
    *   **Comprehensive Theory:** Provides a full suite of rigorous guarantees, including a novel joint regret-coverage bound, essential distribution-free finite-sample coverage, and asymptotic consistency, establishing strong reliability.
    *   **Thorough Experiments:** Validated on a wide range of synthetic and diverse real-world datasets, using multiple base models against a strong set of seven state-of-the-art baselines, convincingly demonstrating its superior performance and general applicability.

*   **Clarity**
    *   The paper is exceptionally well-written, with a clear logical flow that makes complex ideas accessible.
    *   **Excellent Background Explanation:** The background section effectively explains all necessary concepts, providing the context needed to fully appreciate the paper's contribution.

*   **Significance**
    *   Addresses the critical and practical problem of over-conservatism (excessively wide intervals) in online CP, a major barrier to real-world adoption.
    *   The proposed method (COP) is effective, theoretically sound, and simple to implement, giving it high potential to become a standard practitioner's tool.
    *   The theoretical framework introduces new analytical tools that can inspire future research.

**Weaknesses:**

1.  **Lack of Statistical Significance in Experimental Claims:** While the experiments are extensive, the tables present only point estimates (mean and median) for the evaluation metrics from a single run. This makes it difficult to ascertain whether the observed improvements of the proposed method over baselines are statistically significant or merely due to experimental randomness. For example, a small difference in average interval width between two methods might not be meaningful without a measure of variance. The empirical claims would be substantially strengthened by including standard errors or confidence intervals for the reported metrics, which could be derived from multiple runs with different random seeds.

2.  **Over-reliance on Tables for Presenting Results:** The paper relies heavily on large, dense tables (Table 1 and 2) to display the primary experimental outcomes. Such a presentation format makes it challenging for readers to quickly parse results and identify performance trends across different datasets and models. The paper's readability and impact would be greatly enhanced by using visualizations. For instance, it is suggested that the authors consider replacing or supplementing the tables in the main text with figures like **boxplots**. Boxplots would more effectively illustrate the distribution, median, and spread of the prediction interval widths, while also providing a natural way to incorporate the standard error or uncertainty estimates mentioned in the first point, thereby increasing the perceived reliability and clarity of the experimental validation. The detailed tables could then be moved to the Appendix for reference.

**Questions:**

**On the Limits of the CDF Estimator's Robustness:**

A major strength of the paper is that COP maintains coverage guarantees even with an inaccurate CDF estimate. However, a poor estimate could still harm efficiency, potentially making the intervals wider than those from a simpler baseline like OGD.

Could you discuss the practical boundaries of this robustness? Is there a point where a consistently misleading CDF estimate (e.g., during a prolonged, adversarial shift) makes COP perform worse than the baseline OGD in terms of interval width?
An experiment on a synthetic dataset with a deliberately miscalibrated or noisy CDF estimator could help characterize this failure mode and provide practical guidance on when the optimistic step is most beneficial.

**On Statistical Significance of Experimental Results:**

The experimental results in the tables are promising, but they report point estimates from what appears to be a single experimental run. To strengthen the claims of superiority, especially when the performance differences with top baselines are small, could you provide measures of uncertainty for the reported metrics (e.g., standard errors or confidence intervals calculated over multiple runs with different random seeds)? This would be crucial for confirming that the observed improvements are statistically significant.

**On the Impact of Noise and Low Predictability in Scores:**

COP's advantage stems from exploiting predictable patterns in the non-conformity scores. This raises questions about its performance in settings where such patterns are weak or heavily obscured.

How does COP perform in high-noise or low signal-to-noise ratio settings, where the underlying predictable structure in the scores is minimal? In such a scenario, does the optimistic update, which is based on a noisy and potentially uninformative CDF estimate, risk making spurious adjustments that could degrade efficiency (i.e., widen intervals) compared to a more conservative baseline like decay-OGD?
It would be very illuminating to include an experiment on a synthetic dataset where the level of i.i.d. noise added to the non-conformity scores is systematically varied. This would help to characterize the performance trade-offs and define the boundaries of where COP's optimism is a clear advantage.

---

> ### Author Response · Authors · 2025-11-20
> **Response to Reviwer nxNU**
>
> We are sincerely grateful and truly appreciate the time and effort you dedicated under a tight deadline. We have addressed each of your concerns below and are confident that the revised version will be significantly improved thanks to your guidance. Please let us know how we can improve if you still feel we are missing something.
>
> >Strengths: new synthesis of CP with OOGD; comprehensive theory; thorough experiments; effective, theoretically sound, and simple to implement; exceptionally well-written;
>
> Thank you for the summary, and we appreciate your engagement.
>
> >W1: Lack of Statistical Significance in Experimental Claim
>
> A1: We regenerated simulation datasets using ten different random seeds. As shown in the table below, the variability across multiple data generation seeds is small: coverage rate varies by less than 0.5%, and interval width variations vary slightly. This indicates that performance differences between methods are not attributable to any single random dataset. In both experimental settings, COP consistently achieves coverage rates close to the target level while yielding tight interval widths.
>
> We also conducted paired t-tests to reflect the statistical significance of shorter width of COP. All of the baselines above show $p$-values less than 0.05 (e.g., OGD has $p$-value=0.014, SF-OGD has $p$-value=0.0002, and ECI's $p$-value=0.03). These results confirm that the observed differences in width between our methods and the COP baseline are statistically significant.
>
> | Method | Prophet |  |  | AR |  |  | Theta |  |  |
> | :---- | :---- | :---- | :---- | :---- | :---- | :---- | :---- | :---- | :---- |
> |  | Coverage (%) | Average width | Median width | Coverage (%) | Average width | Median width | Coverage (%) | Average width | Median width |
> | ACI | 90.0±0.01 | $\\infty$ | 8.27±0.04 | 90.0±0.01 | $\\infty$ | 8.15±0.04 | 90.0±0.01 | $\\infty$ | 8.33±0.05 |
> | OGD | 90.1±0.02 | 8.63±0.09 | 8.62±0.10 | 90.0±0.02 | 8.46±0.10 | 8.48±0.12 | 90.1±0.03 | 8.68±0.12 | 8.64±0.13 |
> | SF-OGD | 90.0±0.01 | 12.65±0.12 | 11.59±0.12 | 90.0±0.01 | 12.66±0.15 | 11.63±0.15 | 90.0±0.01 | 12.68±0.25 | 11.71±0.28 |
> | decay-OGD | 90.6±0.10 | 8.52±0.20 | 8.34±0.15 | 90.1±0.09 | 8.17±0.22 | 8.11±0.17 | 90.5±0.30 | 8.43±0.30 | 8.29±0.20 |
> | PID | 89.7±0.01 | 11.06±0.15 | 9.36±0.18 | 89.7±0.09 | 10.86±0.10 | 9.18±0.13 | 89.7±0.02 | 10.99±0.20 | 9.32±0.25 |
> | ECI | 89.9±0.01 | 8.32±0.17 | 8.34±0.19 | 89.9±0.09 | 8.22±0.23 | 8.26±0.22 | 89.9±0.02 | 8.38±0.17 | 8.43±0.19 |
> | LQT | 90.4±0.30 | 10.72±1.20 | 9.70±1.00 | 90.0±0.30 | 10.23±1.00 | 9.40±0.90 | 90.2±0.50 | 10.49±0.90 | 9.70±0.95 |
> | COP | 89.8±0.15 | 8.29±0.15 | 8.35±0.15 | 89.4±0.50 | 8.12±0.20 | 8.23±0.20 | 89.8±0.30 | 8.35±0.15 | 8.38±0.18 |
>
> >W2: Over-reliance on Tables for Presenting Results
>
> A2: To enhance readability and clarity, we plot the interval widths for all methods in our added Appendix K, together with zoomed-in panels that highlight periods of rapid market movement. These plots show clear differences in how each method responds to changes in volatility.
>
> >Q1: On the Limits of the CDF Estimator's Robustness:
>
> A3: We have added experimental results of COP with $\\hat{F}\_t$ deliberately corrupted by noise. Please main response (2) for details.
>
> >Q2: On Statistical Significance of Experimental Results:
>
> A4 :  Same as W1.
>
> >Q3：On the Impact of Noise and Low Predictability in Scores:
>
> A3: We have conducted the experiments under three low predictable and extreme situations. Please see our main response (3).

---

> > ### Comment · Reviewer_nxNU · 2025-11-26
> >
> > Thanks for the responses.
> >
> > I noticed that the new tables in the appendix now include measures of statistical significance, which is great. However, I am wondering why these uncertainty measures were not added to the original main tables and were instead presented as a separate experiment. In addition, some of the new appendix experiments do not report statistical significance. Although this may not affect the novelty of the work, it does make the results harder for readers to interpret.
> >
> > Regarding visualization, the new figure is presented separately. A more integrated and intuitive visualization in the main text would make the results easier to understand.
> >
> > The additional experiments do demonstrate the robustness of the proposed method more clearly, which is good.
> >
> > Regarding noise robustness, the work in [1] might be a relevant comparison to include.
> >
> > Regarding the claim of “finite-sample coverage” mentioned in the abstract and introduction, this may be somewhat misleading. The theoretical results in the main text primarily establish long-term coverage, which is the standard guarantee in online conformal prediction, rather than finite-sample coverage in the strict sense.
> >
> > I have updated my rating.
> >
> > ---
> >
> > [1] Xi, Huajun, Kangdao Liu, Hao Zeng, Wenguang Sun, and Hongxin Wei. 2025. “Exploring the Noise Robustness of Online Conformal Prediction.” in The Thirty-ninth Annual Conference on Neural Information Processing Systems.

---

> > > ### Author Response · Authors · 2025-11-26
> > >
> > > Thank you for your careful review and positive feedback. Below is our further response to your comments. We are wondering if you have a new evaluation of our paper after reading our responses. If you have other questions, we are very happy to discuss them with you.
> > >
> > > Regarding statistical significance, we note that most prior online CP methods (including \[2\], \[3\], \[4\], \[5\]) do not typically report such metrics because their update rules are often **deterministic**. To rigorously demonstrate the robustness of our approach, we generated simulated data sequences using 10 different random seeds (sequence length $T=2000$), as detailed in Appendix I. The results show minimal standard deviations across seeds, confirming the stability and reliability of COP. Consequently, the comparisons in the main tables and the added results provide strong evidence of our method's superiority.
> > >
> > > Regarding estimated CDF’s noise robustness, our analysis focuses on **estimation noise** within the auxiliary CDF ($\\hat{F}\_t$) used for the optimistic update, ensuring our method remains valid even when this distributional "hint" is inaccurate or corrupted. In contrast, the work in \[1\] addresses **label noise**, where the observed ground truth $Y\_t$ itself is incorrect (e.g., uniform label noise), and proposes a "Robust Pinball Loss" to correct the resulting gradient bias. COP ensures resilience against misleading distributional estimates, whereas the method in \[1\] ensures resilience against noisy feedback signals.
> > >
> > > Regarding visualization, we sadly put the visualization figures in Appendix K due to space constraints.
> > >
> > > Regarding the theoretical guarantees, we clarify that both Proposition 2 and Theorem 2 in our paper establish the finite-sample coverage guarantees. Their validity holds for any finite time horizon $T$ and does **not rely on asymptotic assumptions ($T \\to \\infty$)**. This focus on finite-sample validity is a key advantage of the ACI-based literature (e.g. \[2\], \[3\], \[4\], \[5\]). Moreover, the finite-sample result in Proposition 2 can naturally imply long-term coverage.
> > >
> > >
> > > \[1\] Xi, H., Liu, K., Zeng, H., Sun, W., & Wei, H. Exploring the Noise Robustness of Online Conformal Prediction. In the Thirty-ninth Annual Conference on Neural Information Processing Systems.
> > >
> > > \[2\] Gibbs, I., & Candes, E. "Adaptive conformal inference under distribution shift." in the Thirty-fourth Annual Conference on Neural Information Processing Systems.
> > >
> > > \[3\] Angelopoulos, A., Candes, E., & Tibshirani, R. J. "Conformal pid control for time series prediction." in the Thirty-sixth Annual Conference on Neural Information Processing Systems.
> > >
> > > \[4\] Angelopoulos, A. N., Barber, R., & Bates, S. Online conformal prediction with decaying step sizes. In the Forty-first International Conference on Machine Learning.
> > >
> > > \[5\] Wu, J., Hu, D., Bao, Y., Xia, S. T., & Zou, C. Error-quantified Conformal Inference for Time Series. In the Thirteenth International Conference on Learning Representations.

---

### Official Review · Reviewer_LFYw · 2025-10-27

**Soundness:** 3
**Presentation:** 3
**Contribution:** 3
**Rating:** 8
**Confidence:** 3

**Summary:**

This paper introduces an online conformal prediction algorithm that incorporates distributional information of non-conformity scores into the update rule. The proposed method leverages an estimated cdf of scores to anticipate predictable patterns and tighten prediction sets while preserving coverage guarantees. By establishing a connection between the proposed method and OOGD, authorss demonstrate a joint bound on regret and coverage.

**Strengths:**

1- The idea of incorporating distribution-informed optimistic updates into online conformal prediction is a meaningful advancement in this area.

2- The paper provides solid theoretical foundations for both coverage and regret.

3-The paper is well written and easy to follow, with clear motivation and smooth storytelling that connects prior work to the proposed method.

**Weaknesses:**

Please check the questions.

**Questions:**

1- How were the hyperparameters chosen for COP?

2- Could the authors include a computational complexity analysis of the proposed method and a comparison with previous baselines, such as ACI?

3- Would authors show empirically whether performance degrades significantly with poor CDF estimation or not?

4- Could the authors clarify whether $q_t$ in line 121  is a type or not? It's not consistent with eq(2)

---

> ### Author Response · Authors · 2025-11-20
> **Response to Reviwer LFYw**
>
> We are sincerely grateful and truly appreciate the time and effort you dedicated under a tight deadline. We have addressed each of your concerns below and are confident that the revised version will be significantly improved thanks to your guidance. Please let us know how we can improve if you still feel we are missing something.
>
> >Q1: How were the hyperparameters chosen for COP?
>
> A1: COP has three hyperparameters, the base learning rate $\\eta$, scale factor $\\lambda=0.5$. To ensure experimental fairness, we set the window $w$ in alignment with PID and ECI. For a specific dataset, the optimal $w$ can be obtained through grid search. As for the bandwidth, we take it to be the same with $w$ to avoid introducing excessive parameters. For fair comparison, same as previous works, the adaptive learning rates $\\eta\_t \= \\eta \\cdot (\\max\\{s\_{t-w+1}, \\dots, s\_t\\} \- \\min\\{s\_{t-w+1}, \\dots, s\_t\\})$.
>
> >Q2: Could the authors include a computational complexity analysis of the proposed method and a comparison with previous baselines, such as ACI?
>
> A2: Yes, definitely\! We have included a detailed computational complexity analysis and reported time cost of per-step updates in main response (1).
>
> >Q3: Would authors show empirically whether performance degrades significantly with poor CDF estimation or not?
>
> A3: Good idea. To further clarify the influence of misspecification, we added experiments about noisy estimated CDF and datasets with low predictability, as can be seen in main response (2) and (3).
>
> >Q4: Could the authors clarify whether q\_t in line 121 is a type or not? It's not consistent with eq(2)
>
> A4: Thank you for your careful reading and we are sorry for the typo. It is the same type and we have revised it.

---

### Official Review · Reviewer_CEXt · 2025-10-30

**Soundness:** 3
**Presentation:** 2
**Contribution:** 2
**Rating:** 4
**Confidence:** 3

**Summary:**

This paper proposes Conformal Optimistic Prediction (COP), an online conformal prediction (CP) algorithm that integrates distributional information of nonconformity scores into the update rule. COP leverages estimated cumulative distribution functions to refine the prediction radius dynamically. A joint coverage–regret bound and finite-sample distribution-free coverage have been established.

**Strengths:**

The paper introduces an elegant refinement step using estimated CDFs. Theoretical contributions include finite-sample and asymptotic coverage guarantees, as well as a joint regret–coverage bound under general learning rates.

**Weaknesses:**

- Although both empirical and kernel-based CDFs are considered, the impact of different estimators or misspecification on performance and validity is not deeply analyzed.


- The presentation could be improved for greater readability. For instance, abbreviations should be used consistently once their full forms have been introduced. Moreover, Sections 3.1–3.3 are mathematically dense, which may make it challenging for readers unfamiliar with OOGD or online conformal prediction to grasp the underlying intuition behind each step.

- Computation time of the methods in experiments.

**Questions:**

- How to choose the scale factor in real practice? How sensitive is COP’s performance to the choice of the scale factor?

- In adversarial settings where estimated CDFs are inaccurate, how robust is COP compared with purely adversarial CP methods like SF-OGD?

- How does COP behave under extreme concept drift or abrupt regime changes, where the empirical CDF is no longer representative of recent data?

---

> ### Author Response · Authors · 2025-11-20
> **Response to Reviwer CEXt**
>
> We are sincerely grateful and truly appreciate the time and effort you dedicated under a tight deadline. We have addressed each of your concerns below and are confident that the revised version will be significantly improved thanks to your guidance. Please let us know how we can improve if you still feel we are missing something.
>
> >W1: Although both empirical and kernel-based CDFs are considered, the impact of different estimators or misspecification on performance and validity is not deeply analyzed.
>
> A1: Thank you for your feedback. Theoretically, our refinement prevents the CDF error from accumulating during the iteration and has little impact on the long-term coverage. To further clarify the influence of misspecification, we added experiments about noisy estimated CDF and datasets with low predictability. Please see main response (2) and (3) for details.
>
> >W2: The presentation could be improved for greater readability. For instance, abbreviations should be used consistently once their full forms have been introduced. Moreover, Sections 3.1–3.3 are mathematically dense, which may make it challenging for readers unfamiliar with OOGD or online conformal prediction to grasp the underlying intuition behind each step.
>
> A2: Thank you for the advice. We have made several revisions to improve the presentation and ensured consistent use of abbreviations throughout the paper (such as the use of OOGD). Regarding the mathematically dense Sections 3.1–3.3, we added more explanatory text to elucidate the underlying intuition (line 172-176), and simplified some of the mathematical expressions for greater readability. Our motivation originates from the following two aspects:
>
> * Previous online CP methods leveraging OCO algorithms primarily address fully adversarial environments. However, in the CP domain, there is often predictable information remaining in the environment.
> * If we assume that the conditional distribution $s_t|S_{t-1}$ is invariant over $t$, the true loss of online CP at time $t$ is actually $\\mathcal{L}\_t(q)=E\_{s\_t} \\ell\_{1-\alpha}(s\_t-q)$. We use estimated CDF to obtain $\\mathcal{L}\_t(q)$ and implicitly leverage the predictable information of scores in this process.
>
> >W3: Computation time of the methods in experiments.
>
> A3: We have reported time cost per-step update. Please see our main response (1) for the results.
>
> >Q1: How to choose the scale factor in real practice? How sensitive is COP’s performance to the choice of the scale factor?
>
> A4: From the traditional view of OOGD, the scale factor should be set as 1 if we know the true distribution function. However, since we use an estimated CDF and it may not be representative of recent data in adversarial environments, taking it to be less than 1 will enhance robustness. The better $\\hat{F}\_t$ fits $F\_t$ (which means the data are more predictable), the higher we take the scale factor to be (see line 249-253).
>
> To avoid hand-tuning, we recommend taking 0.5, which has good performance on all our datasets. This value balances the contribution of the distribution-informed hint and the gradient-based update. Moreover, we have added an Appendix H in the revised version, reporting performance of COP with different scale factors.
>
> >Q2: In adversarial settings where estimated CDFs are inaccurate, how robust is COP compared with purely adversarial CP methods like SF-OGD?
>
> A5: Good question\! We added experimental results of COP with $\\hat{F}\_t$ deliberately corrupted by noise in Appendix G. Please see our main response (2) for the details.
>
> >Q3: How does COP behave under extreme concept drift or abrupt regime changes, where the empirical CDF is no longer representative of recent data?
>
> A3: We have conducted the experiments under three low predictable and extreme situations. Please see our main response (3).

---

> > ### Comment · Reviewer_CEXt · 2025-11-26
> >
> > Thank you for the authors' efforts. My concerns have been addressed, and I have updated my score accordingly.

---

> > > ### Author Response · Authors · 2025-11-26
> > >
> > > Thank you for your careful review and positive feedback. We truly appreciate your recognition of our revisied version!

---

### Official Review · Reviewer_JBdV · 2025-10-30

**Soundness:** 2
**Presentation:** 3
**Contribution:** 2
**Rating:** 4
**Confidence:** 3

**Summary:**

This paper introduces COP, an online conformal method that injects predictable structure via an estimated CDF of nonconformity scores to perform an “optimistic” radius refinement—tightening sets when patterns exist while remaining robust under misspecification. The theory provides a joint coverage–regret bound together with distribution-free, finite-sample coverage under arbitrary learning rates and convergence when scores are i.i.d. Empirically, COP attains target coverage and yields consistently shorter intervals than strong baselines across synthetic shift scenarios and real-world datasets.

**Strengths:**

The paper is original in recasting online conformal prediction as optimistic online gradient descent, blending a standard feedback step with a CDF guided optimistic refinement to reduce conservativeness. The theory is solid, delivering a joint coverage regret bound, distribution free finite sample coverage for arbitrary learning rates, and convergence under i.i.d. scores, while the presentation is clear with precise setup and actionable pseudocode. Experiments across simulated shifts and multiple real world domains show target coverage with consistently tighter intervals, underscoring practical significance and easy deployability, using simple ECDF or KDE plugins.

**Weaknesses:**

The central theory relies on an unverifiable same sign assumption and boundedness that may fail under nonstationary or heavy tailed regimes; an assumption free, non asymptotic refinement bound based on observable CDF error would strengthen the claims. The simulations omit heteroscedasticity, variance changepoints, and heavy tails, lack ablations isolating the optimistic step and ECDF versus KDE, and do not report recovery time or conditional coverage. Practical clarity and scalability are under specified: window and bandwidth choices are sensitive, per step cost appears $(O(w))$ time and $(O(w))$ space without specialized data structures, and the fairness of hyperparameter tuning is unclear; adding adaptive selection rules and explicit complexity with wall clock scaling would improve deployability. Minor exposition gaps remain in mapping the trust region to the step size and in indexing consistency.

**Questions:**

1. Table 2 shows COP attains competitive coverage with narrow intervals, but Proposition 1 hinges on the unverifiable ``same--sign'' assumption.

  (1) Please report how often $\mathrm{sign}(\hat F_{t+1}(\hat q_{t+1}) - (1 - \alpha)) = \mathrm{sign}(F_{t+1}(\hat q_{t+1}) - (1 - \alpha))$ holds on the real-world datasets (e.g., by sliding windows), and correlate this rate with coverage/width. This would clarify whether COP's gains are explained by the assumption or by broader robustness.

  (2) Provide a refinement-step analysis based solely on the observable CDF error
$\varepsilon_t := \sup_q \lvert \hat F_t(q) - F_t(q) \rvert$ that yields
$$
\text{coverage} \;\le\; \Phi(\varepsilon_{1:T}, \{\eta_t\}, \{\lambda_t\})
$$
regardless of the sign, that is, a non-asymptotic worst-case bound that does not rely on the same-sign assumption. For i.i.d. windows, a DKW-type inequality (and for time series, its mixing/martingale analogues) controls $\varepsilon_t$, aligning the strong empirical results with equally robust theory.



2. The simulation studies assume homoscedastic Gaussian noise. To better assess the robustness of COP's CDF-based refinement, please include experiments with heteroscedasticity, variance changepoints, and heavy-tailed noise (e.g., Student-$t$). Please report coverage and width as well as post-shift recovery time (steps to return to target coverage) across these settings, and summarize observed failure modes.


3. For deployability, could you briefly report the time and space complexity of COP’s per-step update (with windowed ECDF/KDE) and a small scaling plot of per-step wall-clock versus window size $w$? If the implementation is $O(w)$ per step (vs. $O(1)$ for OGD/PID), please state this explicitly; if a data structure is used, please indicate the amortized complexity (e.g., $O(\log w)$) and the memory footprint $O(w)$.

---

> ### Author Response · Authors · 2025-11-20
> **Response to Reviwer JBdV [1/2]**
>
> We are sincerely grateful and truly appreciate the time and effort you dedicated under a tight deadline. We have addressed each of your concerns below and are confident that the revised version will be significantly improved thanks to your guidance. Please let us know how we can improve if you still feel we are missing something.
>
> We note that most of the weaknesses are encompassed within the scope of questions. Therefore, we will focus on addressing the questions in our response.
>
> >Q1: Table 2 shows COP attains competitive coverage with narrow intervals, but Proposition 1 hinges on the unverifiable \`\`same-sign'' assumption.
>
> >(1) Please report how often $\\text{sign}(\\hat{F}\_{t+1}(\\hat{q}\_{t+1}) \- (1 \- \\alpha)) \= \\text{sign}(F\_{t+1}(\\hat{q}\_{t+1}) \- (1 \- \\alpha))$ holds on the real-world datasets (e.g., by sliding windows), and correlate this rate with coverage/width. This would clarify whether COP's gains are explained by the assumption or by broader robustness.
>
> >(2) Provide a refinement-step analysis based solely on the observable CDF error $\\varepsilon\_t := \\sup\_q |\\hat{F}\_t(q) \- F\_t(q)|$ that yields coverage; $\\leq \\Phi(\\varepsilon\_{1:T}, \\eta\_t, \\lambda\_t)$ regardless of the sign, that is, a non-asymptotic worst-case bound that does not rely on the same-sign assumption. For i.i.d. windows, a DKW-type inequality (and for time series, its mixing/martingale analogues) controls $\\varepsilon\_t$, aligning the strong empirical results with equally robust theory.
>
> A1:
>
> (1):Since $F\_{t+1}$ is unobservable in the real-world datasets, we have supplemented the manuscript with a small modification (as presented in Appendix A.3) to address this concern. Specifically, we note that $$\[\\hat{F}\_{t+1}(\\hat{q}\_{t+1})-(1-\\alpha)\]\\cdot\\left\[F\_{t+1}(\\hat{q}\_{t+1})-(1-\\alpha)\\right\]\\geq0$$
> follows from $$|\\hat{F}\_{t+1}(\\hat{q}\_{t+1})-(1-\\alpha)|\\geq\\sup\_q |F\_{t+1}(q)-\\hat{F}\_{t+1}(q)|\\triangleq \\epsilon\_{t+1}.$$
> Hence, an intuitive way to avoid the unverifiable assumption is to replace eq.(5) in paper with:
> $$
>      q\_{t+1}=\\hat{q}\_{t+1}-\\lambda\_{t+1} \\mathbf{1}(|\\hat{F}\_{t+1}(\\hat{q}\_{t+1})-(1-\\alpha)|\\geq \\epsilon\_{t+1}) \\left(\\hat{F}\_{t+1}(\\hat{q}\_{t+1})-(1-\\alpha)\\right).
>  $$
>
> For deployability, $\\epsilon\_{t+1}$ can be viewed as a hyperparameter depending on the temporal properties of data and the accuracy of $\\hat{F}\_{t+1}$. In cases like i.i.d. or mixing cases, its magnitude can be approximated through DKW-type inequalities.  However, to maintain the simplicity of our method (and avoid introducing extra hyperparameters), we included this supplementary analysis in the Appendix A.3.
>
> (2): Great point! We are happy to clarify our theoretical results in detail and why we do not leverage the CDF error bound. Combining update eq.(4) and eq.(5) in our paper, we have:
> $$q_{t+1}=q_t+\eta(err_t-\alpha)+\eta(M_t-M_{t+1}),$$
> where $M_t=\lambda_t/\eta \hat{F}(\hat{q_t}-(1-\alpha))$ and $\hat{q}_t+\lambda_t(\hat{F}_t(\hat{q}_t)-(1-\alpha))=q_t$. By induction,
> $$\\text{Coverage}=\\frac{\\sum\_{t=1}^T err\_t-\\alpha}{T}=\\frac{q\_{T+1}-q\_1}{\\eta T}+\\frac{M\_{T+1}-M\_1}{\\eta T}.$$
>
> Given that the score $s\_t$ is uniformly bounded, then $q\_t$ is uniformly bounded as proved in Lemma 2\. Hence, **our refinement prevents the CDF error** **from accumulating** during the iteration and has little impact on the long-term coverage.
>
> While the refinement does not impact the long-term coverage, it does have an influence on "instantaneous" performance measured by the expected quantile loss $\\mathcal{L}(q\_t)$. We believe that the "same-sign" assumption is to some extent a necessary and sufficient condition for the existence of a valid $\\lambda\_t\>0$, which is similar to the necessity of correct direction of gradient descent (see line 705-710). If the "same-sign" assumption fails, it indicates that we are steered in the wrong direction and $q\_{t+1}$ will perform worse than \\(\\hat{q}\_{t+1}\\). To clarify the robustness, we additionally present an upper bound for $\\mathcal{L}(q\_{t+1})-\\mathcal{L}(\\hat{q}\_{t+1})$ correlating with $\\epsilon$ instead of the sign, in Appendix B.1, that is:
> $$\\mathcal{L}(q\_{t+1})-\\mathcal{L}(\\hat{q}\_{t+1})\\leq \\lambda\_{t+1}(1+\\epsilon\_{t+1}+\\frac{L}{2}).$$
>
> Although $q\_{t+1}$ may perform worse than $\\hat{q}\_{t+1}$, the degrade is bounded by $\\lambda\_{t+1}$ and CDF error $\\epsilon\_{t+1}$.
> Since we mention that the CDF error will not accumulate, **the "same-sign" assumption is not a prerequisite for the core validity of COP (i.e. coverage), and more of an explanation for refinement's gains.**

---

> > ### Author Response · Authors · 2025-11-20
> > **Response to Reviwer JBdV [2/2]**
> >
> > >Q2: The simulation studies assume homoscedastic Gaussian noise. To better assess the robustness of COP's CDF-based refinement, please include experiments with heteroscedasticity, variance changepoints, and heavy-tailed noise (e.g., Student-). Please report coverage and width as well as post-shift recovery time (steps to return to target coverage) across these settings, and summarize observed failure modes.
> >
> > A2: Great point\! We have conducted the experiments under heteroscedasticity and heavy-tailed noise, variance changepoints, and extreme distribution drift in the main response (3).
> >
> > We additionally report the recovery time below, which is defined as the earliest time after a changepoint when the sliding-window empirical coverage rate consistently returns to the nominal level for 10 consecutive time steps. More details can be seen in Appendix F.
> >
> > | Method | Prophet |  | AR |  | Theta |  |
> > | :---: | :---: | :---: | :---: | :---: | :---: | :---: |
> > |  | Recovery Time 1 | Recovery Ttime 2 | Recovery Time 1 | Recovery Ttime 2 | Recovery Time 1 | Recovery Ttime 2 |
> > | ACI | 35 | 73 | 50 | 0 | 35 | 66 |
> > | OGD | 12 | 0 | 40 | 0 | 40 | 0 |
> > | SF-OGD | 12 | 0 | 12 | 0 | 12 | 0 |
> > | decay-OGD | 6 | 73 | 153 | 73 | 60 | 73 |
> > | PID | 40 | 0 | 12 | 0 | 40 | 0 |
> > | ECI | 12 | 8 | 40 | 0 | 40 | 0 |
> > | LQT | 156 | 73 | 60 | 73 | 60 | 73 |
> > | COP | 12 | 0 | 40 | 0 | 40 | 0 |
> >
> > >Q3: For deployability, could you briefly report the time and space complexity of COP’s per-step update (with windowed ECDF/KDE) and a small scaling plot of per-step wall-clock versus window size w? If the implementation is O(w) per step (vs. O(1) for OGD/PID), please state this explicitly; if a data structure is used, please indicate the amortized complexity (e.g., O(log(w))) and the memory footprint O(w).
> >
> > A3: Thank you for the feedback. The window size $w$ in COP (set to 100 in all experiments) is a constant independent of sample size. For both ECDF and KDE-based per-step updates, the time complexity is O($w$)—consistent with PID and ECI, because of the adaptive learning rate—and the space complexity is O($w$).
> >
> > Regarding deployability, we have clarified the deployability of COP and reported time cost per-step update. Please see our main response (1) for the results.
> >
> > **Weakness not mentioned in questions:**
> >
> > >The central theory relies on boundedness that may fail under nonstationary or heavy tailed regimes.
> >
> > A4: We understand your concern. In general, the boundedness of scores is a ubiquitous assumption in online CP literature and is the theoretical foundation of the line of the works based on iterations. Without this assumption, even OGD cannot yield a valid long-term coverage.
> >
> > >Practical clarity and scalability are under specified: window and bandwidth choices are sensitive, per step cost appears O(w) time and O(w) space without specialized data structures, and the fairness of hyperparameter tuning is unclear; adding adaptive selection rules and explicit complexity with wall clock scaling would improve deployability.
> >
> > A5: Due to the high diversity of the datasets, there is no universal choice for the window $w$. In our paper, to ensure experimental fairness, we set the window $w$ in alignment with PID and ECI. It is worth noting that even choosing constant $w$, COP still outperforms other methods. For a specific dataset, the optimal $w$ can be obtained through grid search. As for the bandwidth, we take it to be the same with $w$ to avoid introducing excessive parameters.
> >
> > >Minor exposition gaps remain in mapping the trust region to the step size and in indexing consistency.
> >
> > A6: Thank you for your careful reading. We have added clarifications for the relationship between the trust region and the step size (see line 196).

---

### Author Response · Authors · 2025-11-20
**Main Responses**

We are grateful to the engaged reviewers, who took a clear interest in the paper and suggested ways to improve it. Thank you\!

**New Content of Revised Manuscript:**

We have presented a revised version, including discussions on the "same-sign" assumption  (Appendix A.3), experimental results under three other simulation datasets (Appendix E), Post-shift coverage recovery time (Appendix F), robustness of inaccurate estimated CDF (Appendix G), sensitivity analysis of the scale factor (Appendix H), statistical significance analysis (Appendix I), computational complexity analysis (Appendix J), visualization of coverage and interval (Appendix K).
We hope to respond to the critical comments through the following extensive revisions:

**1\. Analysis of Computational Overhead：**

The simplest gradient-based methods (OGD and decay-OGD) are the fastest, requiring only 0.001 ms per step. Methods that consider all past data, such as SF-OGD and PID, incur higher overhead (around 0.012 – 0.013 ms). Methods that consider the past data in a certain window, such as ECI, LQT, and COP fall in a similar range (0.007 – 0.011 ms). Overall, the differences between methods remain small in absolute terms, and all methods run easily in real time for typical streaming applications.

We also observe that as $w$ increases (from 0 to 500), the per-step runtime remains largely stable. Due to the high efficiency of NumPy's vectorized operations, the incremental computational cost associated with $w$ is negligible compared to the fixed overhead of the Python interpreter. To support this, we have added a table comparing COP’s per-step time with other baselines below. It is shown that the slight overhead is well-justified by COP’s superior performance of tighter prediction sets with valid coverage. The details can be seen in Appendix J in our revised version.

| Method | ACI | OGD | SF-OGD | decay-OGD | PID | ECI | LQT | COP |
| :---- | :---- | :---- | :---- | :---- | :---- | :---- | :---- | :---- |
| Time Cost (ms) | 0.043 | 0.001 | 0.012 | 0.001 | 0.013 | 0.007 | 0.010 | 0.011 |

**2. Robustness of Inaccurate Estimated CDF:**

We have added experimental results of COP with $\\hat{F}\_t$ deliberately corrupted by noise. Specifically, at each time step, we construct a noisy ECDF by  $  \\hat{F}\_{noisy} \= \\gamma\\hat{F} \+ (1-\\gamma)\\epsilon,  $
where $\\hat{F}$ is the ECDF, $\\epsilon \\sim U(0,1)$ denotes a uniform random noise, and $\\gamma \\in \\{1, 0.9, 0.5, 0.1, 0\\}$ controls the level of the corruption.

The table below shows that the coverage rate remains stable across all values of $\\gamma$, even when the CDF is fully replaced by uniform noise ($\\gamma=0$). **It coincides with our theoretical results of theorem 2 that COP achieves long-term coverage regardless of misspecification of $\\hat{F}\_t$.** As for the width, we note that COP consistently performs better than SF-OGD when taking $\\gamma=1, 0.9, 0.5$, and exhibits comparable performance when taking $\\gamma=0.1$. Only when the CDF is fully replaced by uniform noise ($\\gamma=0$) does SF-OGD outperform COP. The details can be seen in Appendix G  in our revised version.

| Dataset | γ | Prophet |  |  | AR |  |  | Theta |  |  |
| ----- | ----- | ----- | ----- | ----- | ----- | ----- | ----- | ----- | ----- | ----- |
|  |  | Coverage (%) | Average width | Median width | Coverage (%) | Average width | Median width | Coverage (%) | Average width | Median width |
| Amazon Stock | 1.0 | 89.6 | 39.86 | 27.91 | 89.5 | 17.09 | 12.90 | 89.6 | 17.21 | 12.23 |
|  | 0.9 | 89.6 | 38.95 | 27.85 | 89.3 | 17.14 | 12.71 | 89.6 | 17.45 | 13.12 |
|  | 0.5 | 89.7 | 42.24 | 29.56 | 89.4 | 17.24 | 13.08 | 89.6 | 17.20 | 12.76 |
|  | 0.1 | 90.1 | 46.93 | 30.89 | 89.2 | 17.45 | 12.44 | 89.6 | 17.60 | 12.73 |
|  | 0.0 | 90.0 | 62.65 | 53.09 | 89.7 | 22.07 | 18.26 | 89.5 | 30.76 | 28.56 |
| Google Stock | 1.0 | 89.7 | 49.72 | 42.09 | 89.6 | 19.87 | 17.04 | 89.3 | 30.25 | 28.24 |
|  | 0.9 | 89.8 | 49.99 | 42.06 | 89.6 | 19.81 | 16.92 | 89.4 | 30.43 | 27.79 |
|  | 0.5 | 90.0 | 54.57 | 46.08 | 89.6 | 19.94 | 17.33 | 89.4 | 29.82 | 27.27 |
|  | 0.1 | 90.0 | 61.25 | 51.91 | 89.6 | 19.95 | 17.23 | 89.5 | 30.68 | 28.18 |
|  | 0.0 | 90.8 | 86.26 | 55.43 | 90.0 | 30.04 | 20.32 | 89.9 | 30.24 | 20.76 |

**3\. Impact of Noise and Low Predictability of Scores:**

To better assess the robustness of COP's CDF-based refinement, we added experiments in three other simulation datasets under variance changepoint, heavy-tailed noise, and extreme distribution drift setting, respectively. In the tables below, we reported coverage, width and recovery time across these settings. We note that COP performs comparably to OGD and consistently outperforms other baselines in most cases, indicating the robustness. The details can be seen in Appendix E in our revised version.

---

> ### Author Response · Authors · 2025-11-20
> **Experiments for Low Predictability of Scores**
>
> | Dataset | Method | Prophet |  |  | AR |  |  | Theta |  |  |
> | :---: | :---: | :---: | :---: | :---: | :---: | :---: | :---: | :---: | :---: | :---: |
> |  |  | Coverage | Average width | Median width | Coverage | Average width | Median width | Coverage | Average width | Median width |
> | Variance Changepoint | ACI | 91.0  | $\\infty$ | 11.06  | 91.0  | $\\infty$ | 10.74  | 91.0  | $\\infty$ | 10.90  |
> |  | OGD | 90.1  | 10.57  | 10.65  | 90.0  | 10.43  | 10.25  | 90.0  | 10.44  | 10.07  |
> |  | SF-OGD | 90.0  | 14.75  | 14.06  | 90.0  | 14.80  | 14.18  | 90.0  | 14.47  | 13.82  |
> |  | decay-OGD | 90.2  | 10.71  | 11.21  | 89.9  | 10.41  | 11.00  | 90.2  | 10.76  | 11.12  |
> |  | PID | 89.7  | 13.17  | 11.81  | 89.7  | 13.07  | 11.77  | 89.7  | 13.26  | 11.99  |
> |  | ECI | 89.9  | 10.66  | 10.39  | 89.8  | 10.35  | 9.75  | 89.9  | 10.60  | 10.04  |
> |  | LQT | 90.6  | 12.10  | 11.77  | 88.7  | 10.84  | 10.97  | 89.8  | 11.64  | 11.38  |
> |  | COP | 89.9  | 10.78  | 10.79  | 89.8  | 10.37  | 9.87  | 89.9  | 10.66  | 10.10  |
> | heavy-tailed | ACI | 90.3  | $\\infty$ | 10.03  | 90.2  | $\\infty$ | 10.01  | 90.3  | $\\infty$ | 9.82  |
> |  | OGD | 90.0  | 9.94  | 9.95  | 90.0  | 9.91  | 9.95  | 90.2  | 10.18  | 10.25  |
> |  | SF-OGD | 90.0  | 15.22  | 14.05  | 90.0  | 15.10  | 13.93  | 90.0  | 15.43  | 13.91  |
> |  | decay-OGD | 90.3  | 9.97  | 10.01  | 89.9  | 9.69  | 9.77  | 90.8  | 10.10  | 9.94  |
> |  | PID | 89.7  | 13.60  | 11.47  | 89.6  | 13.14  | 11.36  | 89.6  | 13.12  | 11.19  |
> |  | ECI | 89.9  | 10.54  | 10.49  | 90.0  | 10.34  | 10.27  | 90.0  | 10.55  | 10.36  |
> |  | LQT | 92.0  | 13.21  | 12.32  | 89.4  | 10.13  | 10.09  | 92.8  | 15.79  | 13.74  |
> |  | COP | 89.8  | 9.56  | 9.79  | 89.9  | 10.38  | 10.27  | 90.4  | 9.87  | 10.03  |
> | Extreme Drfit | ACI | 90.4  | $\\infty$ | 79.01  | 89.7  | $\\infty$ | 61.85  | 90.3  | $\\infty$ | 70.06  |
> |  | OGD | 91.1  | 275.87  | 280.00  | 89.9  | 64.03  | 62.00  | 91.3  | 213.10  | 212.00  |
> |  | SF-OGD | 92.0  | 423.68  | 445.98  | 89.9  | 68.23  | 63.62  | 92.4  | 376.03  | 388.03  |
> |  | decay-OGD | 89.7  | 265.81  | 274.93  | 90.0  | 64.50  | 60.32  | 91.6  | 246.51  | 248.24  |
> |  | COP | 91.1  | 275.82  | 279.84  | 89.9  | 64.10  | 62.18  | 91.3  | 213.54  | 211.91  |

---

### Meta-Review · Area_Chair_fy29 · 2026-01-09

**Summary:**

The following key concerns were raised by the reviewers:

1. Limited evidence that the proposed method is robust, including cases where the CDF is inaccurately estimated, noise is present, or scores are highly unpredictable (CEXt, nxNU, LFYw, JBdV)
2. Reliance of theory on same-sign assumption and boundedness (JBdV)
3. Statistical significance of experimental claims and reliance on tables for results presentation (nxNU)

**Reviewer Concerns:**

The authors addressed these concerns in the rebuttal as follows:
1. The authors added experiments exploring the effect of noise on CDF estimation. These results show that coverage is preserved even in the presence of noise. The authors added experiments on three simulated datasets that capture variance changepoints, heavy-tailed noise, and distribution shift. These results show that the proposed method maintains target coverage and produce competitive interval widths relative to baselines.
2. The authors added theoretical results that demonstrate how to circumvent the same-sign assumption. They also clarify that the proposed method prevents CDF error accumulation which enables long-term coverage. These new theoretical results necessitate minor rewriting for more thorough integration into Section 3.1. Importantly, the same-sign assumption is not necessary for the validity of the proposed method but rather an exploration of why the proposed method performs well.
3. The authors added error bars to summarize variability in results across ten seeds. They also conducted paired t-tests to compare the interval widths of the proposed method to baselines.

Therefore, the reviewer’s concerns have been addressed during the rebuttal phase.

**Reviewer Scores:**

- CEXt is very likely to increase their score. The authors clarified that according to their theory the CDF error will not accumulate over iterations. They also added new experiments demonstrating the coverage is preserved in the presence of noise.
- nxNU is very likely to increase their score. Their concerns about statistical significance, robustness, and low predictability were addressed by new results added during the rebuttal phase.
- LFYw is virtually certain to keep their score. Their review was already very positive. The authors provided new experiments showing the effect of poorly estimated CDFs on the results.
- JBdV is likely to increase their score. The authors thoroughly responded to concerns about the same-sign assumption and also provided experiments exploring the effect of heteroskedasticity, variance changepoints, and heavy-tailed noise as requested by the reviewer.

---

### Decision · Program_Chairs · 2026-01-26

Accept (Poster)